# River plastic transport affected by tidal dynamics

Louise J. Schreyers[1], Tim H.M. van Emmerik[1], Thanh-Khiet L. Bui[2], Khoa Van Le Thi[1,3], Bart Vermeulen[1], Hong.-Q. Nguyen[2], Nicholas Wallerstein[1], Remko Uijlenhoet[1,4], and Martine van der Ploeg[1]

[1]Hydrology and Environmental Hydraulics Group, Wageningen University and Research, The Netherlands
[2]Institute for Circular Economy Development, Vietnam National University, Vietnam
[3]Faculty of Water Resources, Hanoi University of Natural Resources and Environment, Vietnam
[4]Department of Water Management, Delft University of Technology, Delft, The Netherlands

**Correspondence:** Louise Schreyers (louise.schreyers@wur.nl)

**Abstract.** Plastic is an emerging pollutant, and the quantities in rivers and oceans are expected to increase. Rivers are assumed to transport land-based plastic into the ocean, and the fluvial and marine transport processes have been relatively well studied to date. However, the processes controlling the transport in tidal rivers and estuaries, the interface between fluvial and marine systems, remain largely unresolved. For this reason, current estimates of riverine plastic pollution and export into the ocean remain highly uncertain. Hydrodynamics in tidal rivers and estuaries are influenced by tides and freshwater discharge. As a consequence, flow velocity direction and magnitude can change diurnally. In turn, this impacts the transport dynamics of solutes and pollutants, including plastics. Plastic transport dynamics in tidal rivers and estuaries remain understudied, yet the available observations suggest that plastics can be retained here for long time periods, especially during periods of low net discharge. Additional factors such as riparian vegetation and riverbank characteristics, in combination with bidirectional flows and varying water levels, can lead to even higher likelihood of long-term retention. Here, we provide a first observation-based estimate of net plastic transport on a daily time scale in tidal rivers. For this purpose, we developed a simple Eulerian approach using sub-hourly observations of plastic transport and discharge during full tidal cycles. We applied our method to the highly polluted Saigon river, Vietnam, throughout six full tidal cycles in May 2022. We show that the net plastic transport is about 20-33% of the total plastic transport. We found that plastic transport and river discharge are positively and significantly correlated (Pearson's $R^2 = 0.76$). The net transport of plastic is higher than the net discharge (20-33% and 16%, respectively), suggesting that plastic transport is governed by other factors than water flow. Such factors include wind, varying plastic concentrations in the water, and entrapment of plastics downstream of the measurement site. The plastic net transport rates alternate between positive (seaward) net transport and negative (landward) net transport, as a result of the diurnal inequality in the tidal cycles. We found that soft and neutrally buoyant items had considerably lower net transport rates than rigid and highly buoyant items (10-16% vs 30-38%), suggesting that transport dynamics strongly depends on item characteristics. Our results demonstrate the crucial role of tidal dynamics and bidirectional flows in plastic transport dynamics. With this paper we emphasize the importance of understanding fundamental transport dynamics in tidal rivers and estuaries to ultimately reduce the uncertainties of plastic emission estimates into the ocean.

# 1 Introduction

Exposure of terrestrial and aquatic ecosystems to plastic has gained considerable interest among the public and scientific community, due to its potential negative effects on the environment (Rochman et al., 2016). While the environmental risks posed by plastics remain to date largely uncertain, its presence in the environment is widely perceived as undesirable from an economic, aesthetic and ethical perspective (Borrelle et al., 2017; Koelmans et al., 2021; Beaumont et al., 2019). Effective and timely reduction strategies require understanding of the transfer dynamics of plastics across ecosystems and within environmental compartments (van Emmerik and Schwarz, 2020). Rivers are one of the main pathways for the delivery of plastics from land to the sea (Meijer et al., 2021). Recently, efforts have been made to use concepts from hydraulics, hydrology, fluvial geomorphology, sedimentology, and debris transport to resolve the open questions of river plastic transport (Liro et al., 2020; Valero et al., 2022; Waldschläger et al., 2022). In particular, river plastic transport processes have been increasingly investigated in recent years in relation to hydrology. Observational studies have demonstrated the strong response of plastic transport to high river discharge events (van Emmerik et al., 2022a, 2023). Extreme discharge events such as floods mobilize large quantities of plastic and can lead to increased plastic emissions into the ocean (Roebroek et al., 2021a; van Emmerik et al., 2023; Hurley et al., 2018). Under normal hydrological conditions, the relation between plastic transport and discharge varies between catchments and is non-trivial (Roebroek et al., 2022; van Emmerik et al., 2022a). Despite growing efforts to link plastic transport to hydrological processes, the transfer dynamics from rivers to sea remain poorly understood (van Emmerik et al., 2022b). Ultimately, the transfer processes in the lower reaches of rivers - in tidal rivers and estuaries - are the most crucial aspect for quantifying plastic emissions into the ocean. Yet, these plastic transfer processes at the river-ocean interface are arguably the most understudied aspect of riverine plastic transport.

Tidal rivers and estuaries are key components of river systems, as they form the interface between rivers and coastal environments (Hoitink and Jay, 2016). In tidal rivers, flows are affected by the combination of freshwater discharge and coastal forcing processes, such as tides. The interactions between river discharge and tidal dynamics ultimately affects the water, sediment and pollutant budgets (Healy et al., 2007; Tessler et al., 2018; Fernandes and Pillay, 2010). This can result in either net export towards the coastal water or net import landward, depending on the spatio-temporal scales considered. For example, characterizing net sediment transport requires quantifying the balance between landward supply and retention mechanisms within the estuarine zone. Various pollutants are similarly affected by bidirectional flows, with both net export and import being observed depending on the tidal dynamics (Fernandes and Pillay, 2010).

Several plastic research studies have aimed to quantify global riverine emissions of plastic into the sea (Jambeck et al., 2015; Lebreton et al., 2017; Schmidt et al., 2017; Meijer et al., 2021). River transport models typically include freshwater discharge as a determining variable for the total export into the sea, but do not consider tidal effects on net water discharge (Lebreton et al., 2017; Schmidt et al., 2017; Meijer et al., 2021). To date, no plastic transport model accounts for the influence of tidal dynamics on plastic emissions into the sea. Meijer et al. (2021) postulated that the probability of riverine plastic reaching the oceans increases with proximity to the river mouth, because larger cross-sectional areas in downstream reaches will reduce the likelihood of plastic trapping along riverbanks. We argue that in tidal rivers and estuaries, bidirectional flows and other

processes including turbulent mixing, entrapment in mudflats and vegetation could generate the opposite effect. With increasing tidal influence towards the river mouth, higher retention times of plastic within the system can be expected. This can ultimately result in lower net plastic transport rates in the downstream reaches than in the upstream reaches of the river system. Most global models assume that river plastic emissions are equivalent to plastic transport estimated at the most downstream point of the river (Meijer et al., 2021). This neglects retention dynamics within tidal rivers and estuaries, as well as potential landward transport. Acha et al. (2003) found that salinity fronts in estuaries act as a physical barrier that accumulates plastic. More recent studies have also shown the limited nature of plastic export in estuaries (Fernandino et al., 2016; López et al., 2020; Tramoy et al., 2020a; Van Emmerik et al., 2020). For instance, López et al. (2020) simulated plastic transport in the Chesapeake estuary (USA) and found that only 5% of the annual microplastic transport was exported into coastal waters, whereas the overwhelming majority (94%) beached on the estuarine shores.

Both Eulerian and Lagrangian-based approaches have been used to study solutes transfer dynamics from rivers to the ocean, notably in the field of sediment transport (Ballio et al., 2018). Lagrangian approaches follow the motion of particles, whereas Eulerian approaches describe the motion of particles over a spatially fixed volume. Most observation-based studies on plastic transport in tidal rivers and estuaries follow a Lagrangian approach, in that they study the transport and accumulation dynamics of a finite number of items (Ledieu et al., 2022; Ryan and Perold, 2021; Sutton et al.; Tramoy et al., 2020a, b). These studies all show that plastic trajectories are affected by both non-uniform advection (longitudinal) and diffusive (multi-directional) transfer processes. Mobile plastics travel limited distances, although a considerable share of plastics will deposit in various riverine compartments and be retained for years to decades at a time (Tramoy et al., 2020b, a). Such transfer dynamics are the result of both limited transport caused by bidirectional flows and (temporary) trapping in vegetation and along riverbanks. Despite the growing evidence that tidal and estuarine dynamics attenuate plastic emissions into the oceans, net plastic transport has never been measured during full tidal cycles. The difficulty in conducting measurements at night (due to the lack of daylight) and the resource intensive nature of continuous measurements likely explain why such measurements have not been done thus far.

For this study, we developed a simple and easily transferable approach to quantify net plastic transport over tidal cycles at a river cross-section, in relation to total plastic transport. By using a Eulerian approach, we considered a fixed spatial domain in which we estimated plastic transport. This approach entails measuring plastic transport and water flow dynamics (river discharge, flow velocity and water depths) at a sub-hourly frequency. We applied this method to the Saigon river, Vietnam, in May 2022, and estimated net and total plastic transport over six full tidal cycles. For the first time, we were able to estimate net plastic transport in a tidal river, based upon field observations and using an Eulerian approach. We collected data on floating plastic transport for various plastic types and measured river discharge at a sub-hourly frequency. We estimate net transport of plastic and how it varies by plastic type and by tidal cycle. We aim to highlight the crucial role of tidal rivers in the transport of riverine plastic into the ocean.

## 2 Methods

### 2.1 Study site

The field measurements were conducted at one site on the Saigon river (Vietnam), in Ho Chi Minh City (HCMC), at 70 km from the river mouth (Fig.1). The Saigon river originates in Cambodia, passes through the Dau Tieng reservoir, progresses through a diverse agricultural and industrial region and then crosses HCMC, which has a population of 9 million people, making it Vietnam's largest city. Approximately 20 km south of HCMC, the Saigon river meets the Dong Nai river where it forms the Nha Be river. The latter passes through the Can Gio Mangrove forest where it branches into multiple channels and then debouches into the East Sea (Nguyen et al., 2020) (Fig.1A). The Saigon river is affected by an asymmetric semi-diurnal tidal regime, usually resulting in a reversal of the flow direction twice a day. Tidal dynamics are registered up to the Dau Tieng reservoir, 140 km from the river month (Nguyen et al., 2021), and this regulates net discharge in the Saigon river (Camenen et al., 2021). In addition, river discharge in the Saigon river is affected by a strong seasonality between the wet and the dry seasons, with monthly mean net discharge varying between $-80$ and $320$ m$^3$/s (Camenen et al., 2021).

### 2.2 Measurement setup

This study focuses on the transport of floating macroplastics larger than 0.5 cm, hereafter referred to as plastic. We measured plastic transport, water depth, and flow velocity at the Thu Thiem bridge (10.785984, 106.718332), located in the southern part of HCMC. The field measurements were conducted continuously over 74 hours and 30 minutes, from 1 to 4 May, 2022. Five observation points were monitored across the river width, to account for the spatial variability at the river cross-section in plastic transport, water depth, and flow velocity (Fig.1B). The observation points were chosen in order to maximize coverage of the entire river cross-section on the one hand and to minimize the influence of the bridge piers that support the road from which observations were made. Measurements were conducted on both sides of the bridge. During flood flow, the measurements took place on the northern side of the bridge, while the southern side was used during ebb flow. This allowed surveyors to face the flow direction during measurements and facilitated the handling of equipment in and out of the water. The bridge deck was approximately 14 m above the average water surface elevation during measurements.

At each measurement location three instantaneous measurements were taken: floating plastic transport (section 2.3), the water depth (section 2.4), and the flow velocity (section 2.5). A minimum of two surveyors were present to conduct the instantaneous measurements. This was necessary during peak plastic transport periods, when values of up to over 100 items/min were registered. In such cases, one surveyor conducted the visual counting while another noted down the values. Up to four surveyors could be present for instantaneous measurements, depending on availability. Each measurement round lasted on average 9 minutes. The measurement duration varied between 3 to 42 minutes, depending on the number of available surveyors, the presence of boat traffic which could further delay the measurement, and potential challenges with handling equipment. Measurements were conducted both during the day and at night. At night, a flashlight lamp (P18R Signature, Ledlenser, Germany, https://ledlenser.com/en/) was used to illuminate the water surface, estimate plastic transport and take equipment in and out of the water safely. The model used had a 4500 lumen luminous flux.

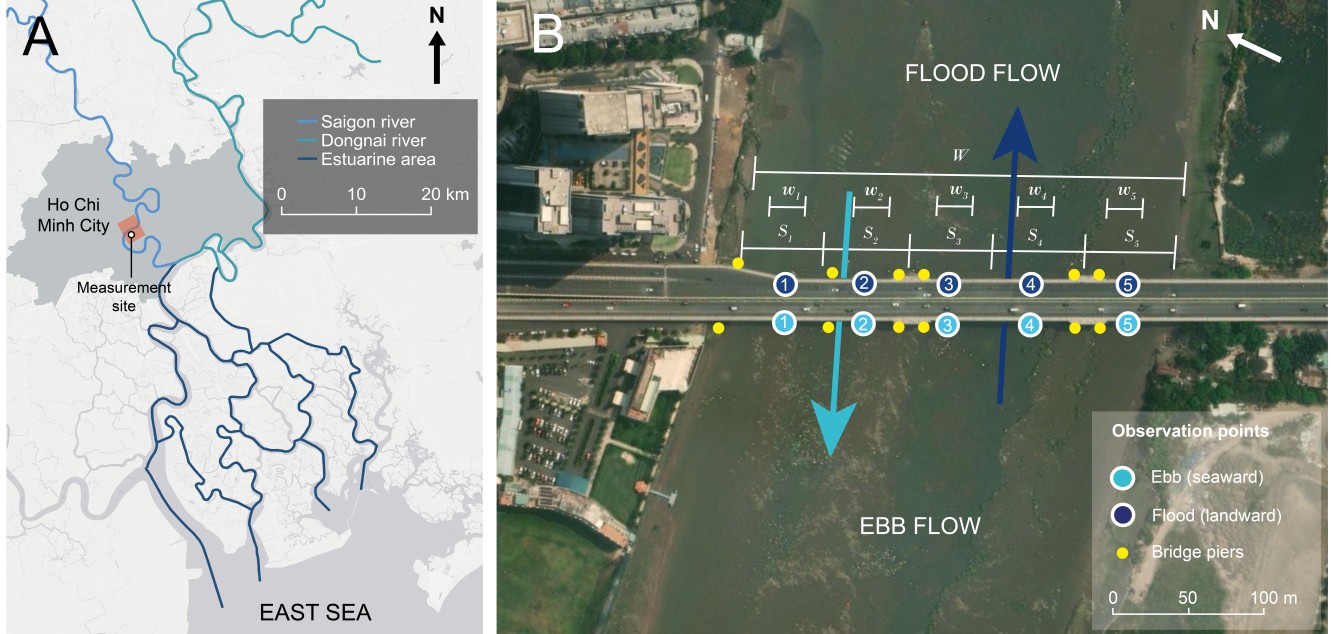

**Figure 1.** A. Measurement site within the Saigon - Dong Nai river system. B. Measurement site (Thu Thiem bridge, 10.785984, 106.718332) and locations. The numbers 1, 2, 3... mark the observation points distributed across the bridge, with variations in their location depending on the flow direction. For floating plastic, we considered observational track width $w_i$ (of 15 meters). For discharge calculations, we considered widths represented as $s_i$ at each observation point. Copyright: Bing Maps. Note the different north orientation for the two panels.

## 2.3 Plastic transport estimates

Plastic transport was estimated using the visual counting method, developed by González-Fernández and Hanke (2017). All visible ($> 0.5$ cm) anthropogenic litter items floating at the water surface were counted and classified for a duration of 2 minutes by trained observers who stand on a bridge. The following eight categories were used for classifying visible anthropogenic litter items: EPS (expanded polystyrene), $PO_{hard}$ (hard polyolefins), $PO_{soft}$ (soft polyolefins), PS (polystyrene), PET (polyethylene terephthalate), Multilayer plastics, Other plastic items and Other litter items (non-plastic). These plastic categories have been used in previous studies (van Emmerik et al., 2022a; Schreyers et al., 2021) and are considered suitable for a first-order identification of plastic types. In this study, we only consider plastic items and therefore do not report total litter transport estimates. Plastic transport $P$ [items/hour] was calculated using the following equation (van Emmerik et al., 2022a):

$$P = \frac{W}{5} \sum_{i=1}^{5} \frac{\overline{p_i}}{w_i} \tag{1}$$

With mean plastic transport observation $\overline{p_i}$ [items/hour] for observation point $i$ at 5 observation points, observation track width $w_i$ [m] and total river width $W$ [m]. We considered an observation track width of 15 m, and a total river width of 298 m.

Plastic transport is often expressed in terms of mass transport in current literature (Lebreton et al., 2017; Meijer et al., 2021; Schmidt et al., 2017; van Emmerik et al., 2022a). Therefore, we also expressed plastic transport $M$ in terms of mass transport [kg/d], using the following equation (Vriend et al., 2020):

$$M = P \cdot \overline{m} \cdot c \tag{2}$$

With $\overline{m}$ expressing either the mean or median mass per plastic item [g] and $c$ the conversion factor from g/hour to kg/d. To convert item transport to mass transport, we used the mass statistics from van Emmerik et al. (2019). In this study, 3,022 items collected over 45 days were weighted and categorized into the following plastic type categories: EPS, PS, $PO_{hard}$, $PO_{soft}$ and PET. The items were collected using a net at the same monitoring location as in our study. The weighting and counting of individual items reported in (van Emmerik et al., 2019) allowed us to derive mean and median masses per item category. For the categories 'Multilayer' and 'Other plastic' from our observations, we used the mean and median mass found for '$PO_{soft}$' items and all items, respectively. In van Emmerik et al. (2019), 'Multilayer' items were categorized as soft items ($PO_{soft}$). Median and mean mass values per item category are reported in Table D3 (Appendix).

## 2.4 Water depth, flow velocity measurements and discharge estimates

Water depth was measured using a single beam sonar with Compressed High Intensity Radiated Pulse (CHIRP) (Deeper Smart Sonar Chirp 2, Lithuania, https://deepersonar.com/). The sonar was lowered from the bridge into the water using a rope. Once the sonar reached the water surface, water depth values could be read on a previously paired mobile phone using the Deeper Smart Sonar mobile application. The sonar was lost on 4 May, 2022 around 03:00 A.M. due to collision with a container ship. As a result, water depths were not recorded for the last 13 hours of measurements.

Near-surface flow velocities were measured using a propeller flow meter (Flowatch, JDC, Switzerland, https://www.jdc.ch/). The flow meter was lowered from the bridge into the water, at approximately one meter of depth from the surface, using a cable. The surface velocities were converted to depth-average velocity by multiplying the surface velocity by a coefficient of 0.85. This coefficient assumes a logarithmic vertical velocity distribution and a typical bed roughness and is generally accepted in the hydrological community (Muste et al., 2008; Hauet et al., 2018; Rantz, 1982; Boiten, 2003). Flow velocities for flood water flows were recorded as negative values, and as positive values for ebb water flows.

The cross-sectional area was estimated for each observation point, as follows:

$$a_i = s_i \cdot d_i \tag{3}$$

With segment width $s_i$ [m] and depth $d_i$ [m] per observation point $i$. We established five segments, with an observation point in the middle. The water depth was measured at each observation point $i$ and was considered as the averaged depth per segment. We estimated water discharge [m$^3$/s] at the river cross-section as follows:

$$Q = \sum_{j=1}^{5} a_i \cdot \overline{v_i} \tag{4}$$

With $\overline{v_i}$ the depth-averaged flow velocity [m/s] at each measurement location $i$. Bathymetric data was not available, and our estimates of water depths could have overlooked local bed variations and scour holes. We measured water depths at five locations across the river width, taking into account contraction scour effects (Arneson, 2013). However, we did not directly measure water depths at the nose of the bridge piers, which could mean that we may have overlooked local scour holes. We estimated the maximum local scour hole depths based on Arneson (2013) (Chapter 7, specifically detailed equations 7.3 and 7.4). We found scour depths reaching maximum values between 3.6-2.7 m, depending on the bridge pier considered (Fig. 1B). We assumed the piers to be composed of a set of two columns on each side of the bridge. We found a maximum total scour area across the entire cross-section of approximately 90 m$^2$. Such increase in cross-sectional area would result in an increase in river discharge estimates by 2%. Thus, we can reasonably assume that under such worst-case scenario, factors such as local scour holes have only a minimal impact on our discharge estimates.

Because of the lack of water depth observations during the last 13 hours of measurement, the resulting discharge estimates only covered 5 out of the 6 tidal cycles. This data gap was filled by estimating river discharge based on the significant and strong relation found with flow velocity for all observed values (Pearson's R$^2$ = 0.99, and p-value < 0.01) (Fig. A1 in Appendix A). The following equation was used to fill missing discharge estimates:

$$Q = \overline{v} \cdot 3325 \tag{5}$$

In addition, it should be noted that precise quantification of discharge was outside the scope for our study. Because of the uncertainties inherent to our discharge estimates, we prefer to report the relationship between plastic transport and water flow based on flow velocity estimates (for instance for Fig.3 and 4).

## 2.5 Temporal data harmonization

Plastic transport, water depths and flow velocities could not be measured at precisely regular time intervals, due to constraints in handling equipment, varying number of available surveyors and varying distances between measurement locations. For this reason, plastic transport, flow velocity and discharge values were interpolated to a regular time interval using two different methods. Flow velocity and discharge values were interpolated using tidal characteristics. Tidal constituents were analyzed using the Unified Tidal Analysis and Prediction (UTide) package in Python 3.4 (Codiga, 2011). This enabled us to determine the coefficients (phase and amplitude) for each tidal constituent, which were in turn used to interpolate our time-series. We present the results of the tidal constituent analysis in Appendix B, as they are not considered novel findings but were nonetheless crucial for flow velocity and discharge interpolation. The temporal interpolation was done to a 5-minute interval. Plastic transport was also interpolated to 5-minute intervals, using a linear interpolation.

## 2.6 Calculating net and total plastic transport and discharge

Here we define ebb and flood as the tidal phases in which the water current is flowing seaward and landward, respectively. While usually seaward plastic transport dominates during the ebb phase and landward plastic transport during the flood phase, short lags in time (of about a few minutes) were noted during slack periods (Fig.2). For instance, although the overall river cross-

section were dominated by one flow direction, reverse flow could still be (temporally) observed at one or a few measurement locations. If at those measurement locations plastic densities out-weigh densities at the remaining measurement locations, a discrepancy can be noted at the cross-section between water flow and plastic transport directions.

     Based on the distinction between flood and ebb phases, we calculated the net plastic transport during ebb and flood, flow velocities and river discharges. We introduce a relative measure of net transport, hereby called the delivery ratio ($d_r$). Using

a relative metric allows for easier comparison across various spatio-temporal scales and within systems with varying plastic pollution levels. The $d_r$ expresses the ratio [-] between net and total transported amounts/volumes/distances, as follows:

$$d_r = \frac{\overbrace{V_e + V_f}^{\substack{\text{Net transported amounts/}\\ \text{volumes/distances } (V_n)}}}{\underbrace{V_e - V_f}_{\substack{\text{Total transported amounts/}\\ \text{volumes/distances } (V_t)}}} \tag{6}$$

     We present two alternative ways of calculating the delivery ratio in Appendix D. The results based on the three ways of calculating $d_r$ are also reported in Appendix D (Tables D1 and D2). For brevity, we only report in the main manuscript the

delivery ratio values as presented above.

     To estimate the delivery ratio ($d_r$), we calculated the total transported amounts, volumes and distances of plastic, discharge and flow velocity during ebb and flood, as follows:

$$V_e = \int_{T_e} f(t)dt \qquad \text{with } \overline{v} > 0 \tag{7}$$

$$V_f = \int_{T_f} f(t)dt \qquad \text{with } \overline{v} < 0 \tag{8}$$

$T_e$ and $T_f$ indicate the ebb and flood tidal phase, respectively, $f$ the values integrated over time $t$ (plastic transport, flow velocity and discharge) and $v$ the flow velocity. The integral values for flow velocity and discharge correspond respectively to the total river distance [m] and water volume [m$^3$] that passed by the measurement location per tidal phase. The integral values for plastic transport correspond to the total amount (number and mass) of plastic items passing by the measurement location. Figure 2

gives an example for the $V_e$ and $V_f$ calculation, using the flow velocity as the variable of reference for distinguishing between flood and ebb.

     We also determined the net plastic transport, flow velocity and discharge ($f_n$) in absolute values (respectively in items/hour, m/s and m$^3$/s) as follows:

$$f_n = \frac{V_e + V_f}{T_e + T_f} \tag{9}$$

In addition, we calculated the mean plastic transport, flow velocity and discharge for each ebb and flood cycle ($f_e$ and $f_f$, respectively), as follows:

$$f_e = \frac{V_e}{T_e} \tag{10}$$

$$f_f = \frac{V_f}{T_f} \tag{11}$$

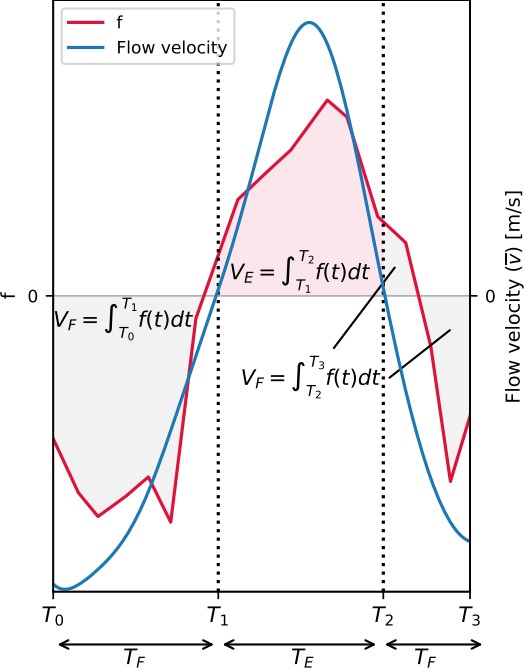

**Figure 2.** Example of calculation of integral areas for the ebb and flood phases of the tidal cycle. The grey shaded areas correspond to the integral during flood, the red shaded area to the integral during ebb. f represents the variable to be integrated, which could be plastic transport, river discharge or flow velocity.

## 3 Results

### 3.1 Net plastic transport less than one-third of total plastic transport

Over the six tidal cycles considered, we found a seaward mean net transport of approximately $2.7 \cdot 10^3$ items/hour, corresponding to 350-470 plastic kg/d (Table 1). This represents only about 20-33% of total plastic transport. This ratio is lower

for river discharge and flow velocity (16%) (Table 1). In the Discussion, we explored potential explanations for the observed
disparities between water and plastic delivery ratios. Plastic net mass transport rates vary by 34%, depending on whether the
mean or median mass of items is considered (Table 1). Overall, the delivery ratio based on mean mass per item is lower (20%)
than that based on median mass per item (33%). We consider the delivery ratio for item transport (25%) to be the more robust
one, as it aligns more closely with our observational data (section 2.3).

**Table 1.** Summary of plastic transport, flow velocity, discharge, and associated metrics during ebb and flood phases.

| | $f_e$ | $f_f$ | $f_n$ | | $V_e$ | $V_F$ | $V_n$ | $d_r$ [-] |
|---|---|---|---|---|---|---|---|---|
| Mass transport (median mass) [kg/d] | $1.4 \cdot 10^3$ | $-7.2 \cdot 10^2$ | $3.5 \cdot 10^2$ | Mass (median mass per item) [kg] | $2.2 \cdot 10^3$ | $-1.1 \cdot 10^3$ | $1.1 \cdot 10^3$ | 0.33 |
| Mass transport (mean mass) [kg/d] | $2.8 \cdot 10^3$ | $-1.9 \cdot 10^3$ | $4.7 \cdot 10^2$ | Mass (mean mass per item) [kg] | $4.4 \cdot 10^3$ | $-2.9 \cdot 10^3$ | $1.5 \cdot 10^3$ | 0.20 |
| Item transport [items/hour] | $1.3 \cdot 10^4$ | $-8.0 \cdot 10^3$ | $2.7 \cdot 10^3$ | Number of items [items] | $5.0 \cdot 10^5$ | $-3.0 \cdot 10^5$ | $2.0 \cdot 10^5$ | 0.25 |
| River discharge [m³/s] | $1.1 \cdot 10^3$ | $-8.1 \cdot 10^2$ | $1.6 \cdot 10^2$ | Water volume [m³] | $1.5 \cdot 10^8$ | $-1.1 \cdot 10^8$ | $4.3 \cdot 10^7$ | 0.16 |
| Flow velocity [m/s] | 0.3 | −0.2 | > 0.0 | Distance [m] | $4.5 \cdot 10^4$ | $-3.2 \cdot 10^4$ | $1.3 \cdot 10^4$ | 0.16 |

Water flow in the Saigon river follows a sinusoidal pattern, with clear alternations between ebb and flood phases determined
by the tidal cycle and its various phases in rising and falling limbs and slack water periods (Fig. 3). The maximum flow velocity
during the ebb phase exceeds that observed during the flood phase (0.6 and −0.4 m/s, respectively). The flood phase is longer
than the ebb phase (38 hours and 20 minutes and 36 hours and 10 minutes, respectively). We found a seaward net discharge of
160 m³/s over the measurement period, corresponding to relative net water transport of approximately 16% of total water flow
(Table 1). Plastic transport follows a similar asymmetrical sinusoidal pattern to flow velocity (Fig. 3). Plastic transport was found
to be highly positively correlated with river discharge and flow velocity (Pearson's $R^2$ = 0.76, and p-value < 0.01 for plastic
transport in relation to both discharge and flow velocity). Plastic transport can be expressed as a linear function in relation to
discharge for all items aggregated (Appendix C, Fig. C1), as well as by plastic types (Appendix C, Fig. C2). For the latter, the
$R^2$ values could indicate the degree to which river discharge influences the transport of these different plastic types. With this
assumption, transport of PS and PO$_{soft}$ items are the most correlated river discharge ($R^2$ of respectively 0.70 and 0.68).
Despite the strong and significant correlation found between river discharge and plastic transport, similar discharge values
were observed for a wide range of plastic transport. For instance, for peak discharges of over 1,800 m³/s, plastic transport varied
by a factor of almost four, between $0.7$-$2.6 \cdot 10^4$ items/hour (Appendix C, Fig. C1). We hypothesize that varying contributions
of different plastic types to the overall plastic transport explain this discrepancy. In particular, a higher share of EPS and PO$_{soft}$,
two types of items for which the relation between transport and river discharge is characterized by a steeper slope (Appendix
C, Fig. C2), might lead to higher transport during peak discharge periods. This hypothesis seems to be confirmed by our
observations (Appendix C, Fig. C1), with EPS and PO$_{soft}$ items making up for more than 80% of the plastic composition
during peak plastic transport, much higher than on average (56%) (Appendix C, Fig. C1). In addition, a hysteresis pattern is
noticeable between plastic transport and river discharges, but was not found to be consistent between rising and falling limbs
of the tidal cycle, for both the entire time-series and across the different tidal cycles observed (Appendix C, Fig. C1 and C3).
Overall, estimating plastic transport based on a simple linear model from measured discharge would yield large uncertainties,

especially for peak transport values. There is no clear explanation for the wide range of plastic transport values during peak discharge events. The observed hysteresis pattern could be related to the asymmetry in rising and falling limb and/or from other sources of uncertainties, including varying concentrations of different plastic types.

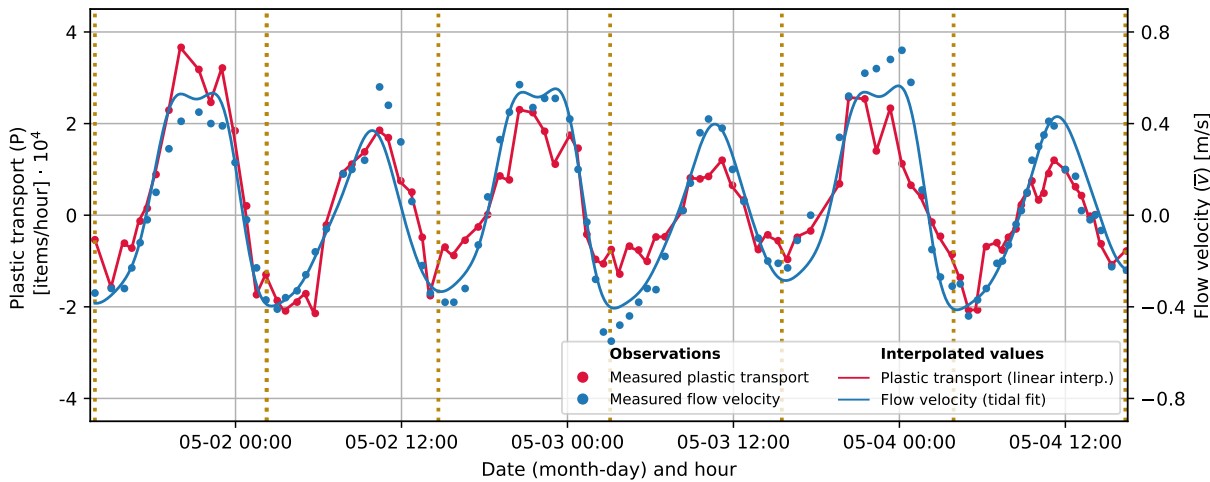

**Figure 3.** Plastic transport and flow velocity over the entire measurement period. The dotted yellow lines separate each tidal cycle.

## 3.2 Diurnal inequality results in alternating positive and negative delivery ratios

During the measurement period, water flow exhibited a mixed tidal cycle (i.e. two high and low tides each lunar day), resulting in diurnal inequality and an alternation between ebb and flood dominated tidal cycles. The first, third and fifth tidal cycles were ebb dominated, as the total volume of water was larger during the ebb phase of the cycle than during the flood phase ($V_e > V_f$). The second, fourth and sixth tidal cycle exhibited, on the contrary, flood dominance ($V_f > V_e$) (Table D2, Appendix D).

     Because of this diurnal alternation, we could therefore expect varying net discharge and plastic transport rates depending
on whether the tidal cycle was ebb or flood dominated. We found positive net plastic transport, flow velocity and discharge, for ebb dominated cycles (1, 3 and 5), for both mean values and delivery ratios (Table 2). Negative net plastic transport, flow velocity and river discharge were measured for flood dominated cycles (2, 4 and 6). This indicates that diurnal variations in tidal dynamics and freshwater discharge, resulting in asymmetry in peaks, are important components in explaining the variability in net flow and transport. In line with this, the tidal constituent analysis showed that the main daily tidal component (K1) is the
second most important tidal component of our time-series (Appendix B, Tidal constituent analysis). As a result of the alternation between ebb and flood dominated cycles, the cycle-averaged net transport rates varied by a factor of nearly 4 between cycles ($9.7 \cdot 10^3$ items/hour for the first cycle and $-2.7 \cdot 10^3$ items/hour for the sixth cycle).

     We hypothesize that high plastic delivery ratios could be governed by either cycle-averaged high net river discharge, high plastic concentrations in the water or a combination of both. For the first tidal cycle, the high plastic delivery ratio (57-63%)

seems to be mainly driven by high plastic concentrations, as the flow velocity and river discharge delivery ratio was not particularly high (32%). The highest mean plastic transport during the ebb phase was found for this cycle ($2.4 \cdot 10^4$ items/hour), almost twice more than for the entire measurement period) (Table D1, Appendix D). For the third tidal cycle, the plastic delivery ratio was closer to the flow velocity and river discharge delivery ratio (40-50% and 39%, respectively), and the net river discharge was found to be quite high (430 m$^3$/s); more than 16% higher in fact than on the first cycle (370 m$^3$/s). This suggests that the high delivery ratio of plastic transport found for the third tidal cycle was mainly governed by high net discharge. The highest plastic delivery ratio was registered during the fifth tidal cycle (59-66%). Net river discharge was also at its peak during this tidal cycle (490 m$^3$/s) and net plastic transport was double the average ($6.7 \cdot 10^3$ items/hour for the fifth tidal cycle and $2.7 \cdot 10^3$ items/hour on average for the entire measurement period, but lower than during the first tidal cycle ($9.7 \cdot 10^3$ items/hour). During the fifth tidal cycle, a combination of high net discharge and high plastic concentrations likely explains the high plastic delivery ratio found.

Plastic delivery ratios calculated based on items transport and mass transport show variable agreement. For ebb dominated cycles, the spread in plastic delivery ratios was comprised between ±1% and ±6%, showing a relatively narrow spread between the calculated values (item transport, mass transport based on median, and mean mass per item). During flood dominated cycles, the spread widens, ranging from ±13% to ±17% (Table 2). This disparity is primarily attributed to lower delivery ratios observed during flood-dominated cycles when considering mass transport based on the median mass per item. The mean mass per item was very similar among items compared to the mean mass of all items: with the exception of PET (mean mass: 20.0 g) all items have a mass comprised between 7.0 and 12.3 g, with an overall average of 10.1 g per item. The median mass was more variable among items, ranging between 1.9 and 7.7 g (with the exception of PET, median mass = 20.8 g) (Table D3). As a result, peaks in transport of items heavier or lighter than others can alter the cycle-averaged net transport rates and delivery ratios. Anticipating on section 3.3, the peak in polystyrene items (PS) observed during the ebb phase of the tidal cycle, can explain the lower delivery ratio registered for the median mass transport. Indeed, the median mass for PS items was higher than the averaged median mass for all items (6.0 g vs 4.3 g, 28% difference), whereas this difference was less pronounced for the mean mass (10.7 g vs 10.1 g, difference of less than 6%).

**Table 2.** Plastic transport, flow velocity, discharge, and associated metrics during ebb and flood phases, per tidal cycle.

| | Cycle | 1 | 2 | 3 | 4 | 5 | 6 |
|---|---|---|---|---|---|---|---|
| $f_n$ | Item transport | $9.7 \cdot 10^3$ | $-1.8 \cdot 10^3$ | $5.3 \cdot 10^3$ | $-9.7 \cdot 10^2$ | $6.7 \cdot 10^3$ | $-2.7 \cdot 10^3$ |
| | Mass transport (median mass) [kg/d] | $1.2 \cdot 10^3$ | $-5.0 \cdot 10^1$ | $5.1 \cdot 10^2$ | $-5.6 \cdot 10^1$ | $6.6 \cdot 10^2$ | $-2.0 \cdot 10^2$ |
| | Mass transport (mean mass) [kg/d] | $2.3 \cdot 10^3$ | $-5.6 \cdot 10^2$ | $8.6 \cdot 10^2$ | $-3.3 \cdot 10^1$ | $1.3 \cdot 10^3$ | $-7.0 \cdot 10^2$ |
| | River discharge [m$^3$/s] | $3.7 \cdot 10^2$ | $-1.9 \cdot 10^2$ | $4.3 \cdot 10^2$ | $-1.2 \cdot 10^2$ | $4.9 \cdot 10^2$ | $-3.3 \cdot 10^1$ |
| | Flow velocity [m/s] | 0.1 | $-0.1$ | 0.1 | $< 0.0$ | 0.2 | $< 0.0$ |
| $d_r$ | Item transport | 0.57 | $-0.15$ | 0.49 | $-0.15$ | 0.66 | $-0.34$ |
| | Mass transport (median mass) | 0.63 | $-0.05$ | 0.50 | $-0.09$ | 0.62 | $-0.27$ |
| | Mass transport (mean mass) | 0.57 | $-0.22$ | 0.40 | $-0.22$ | 0.59 | $-0.40$ |
| | Flow velocity/river discharge | 0.32 | $-0.25$ | 0.39 | $-0.15$ | 0.44 | $-0.04$ |

### 3.3 Net plastic transport varies with plastic type

We determined the transport and delivery ratio per plastic type (Fig. 4). Plastic items differ in their shape, size, buoyancy and rigidity, characteristics that could influence their transport processes. We found that the amplitude in plastic transport varies significantly depending on both the tidal cycle and the type of items considered. Net transport vary by two orders of magnitude depending on the plastic type considered (from $1.3 \cdot 10^3$ items/hour for EPS items to $-3.5 \cdot 10^1$ items/hour for Other plastic items) (Table 3). We calculated a positive net transport in relation to total transport ($d_r > 0$) for all plastic types, with the exception of $PO_{hard}$ and Other plastic. These two categories correspond to the least commonly found items (respectively 3 and 2% of the total plastic items). The delivery ratio varied between 62% and $-16$% depending on the plastic type. Large items such as PET (e.g.: plastic bottles) and rigid and highly buoyant items such as EPS (e.g. expanded polystyrene such as foam) and PS (polystyrene, such as plates) registered the highest net export (62%, 38% and 30%, respectively). On the contrary, soft and neutrally buoyant items such as $PO_{soft}$ (e.g.: bags and foils) and Multilayer (food packaging) had lower delivery ratios (16% and 10%, respectively).

Moreover, large fluctuations in plastic transport were noted depending on the tidal cycle. For instance, transport in EPS, $PO_{soft}$ and PS were particularly high during the first tidal cycle during its ebb phase. Transport of Multilayer items was high during the second tidal cycle, similarly to transport of PS items, also during the ebb phase. Our results suggest that the relative contribution of item types is highly variable, with varying concentrations per plastic type at the water surface, probably resulting from varying inputs of plastics into the river.

**Table 3.** Net plastic transport and delivery ratios per item category. The discrepancy in sign for certain values between net transport and delivery ratios is due to the fact that the latter was calculated based on the integral values for ebb and flood phases, whereas net transport resulted from the difference between mean ebb and flood transport rates.

| Plastic type | Variables | $f_e$ | $f_f$ | $f_n$ | $d_r$ [-] |
|---|---|---|---|---|---|
| EPS | Item transport [items/hour] | $4.8 \cdot 10^3$ | $-2.2 \cdot 10^3$ | $1.3 \cdot 10^3$ | 0.38 |
| | Mass transport (median mass) [kg/d] | $2.2 \cdot 10^2$ | $-1.0 \cdot 10^2$ | $6.1 \cdot 10^1$ | |
| | Mass transport (mean mass) [kg/d] | $8.1 \cdot 10^2$ | $-3.7 \cdot 10^2$ | $2.2 \cdot 10^2$ | |
| $PO_{soft}$ | Item transport [items/hour] | $3.3 \cdot 10^3$ | $-2.4 \cdot 10^3$ | $4.6 \cdot 10^2$ | 0.16 |
| | Mass transport (median mass) [kg/d] | $2.3 \cdot 10^2$ | $-1.6 \cdot 10^2$ | $3.2 \cdot 10^1$ | |
| | Mass transport (mean mass) [kg/d] | $8.3 \cdot 10^2$ | $-6.0 \cdot 10^2$ | $1.2 \cdot 10^2$ | |
| Multilayer | Item transport [items/hour] | $1.9 \cdot 10^3$ | $-1.6 \cdot 10^3$ | $1.7 \cdot 10^2$ | 0.10 |
| | Mass transport (median mass) [kg/d] | $1.3 \cdot 10^2$ | $-1.1 \cdot 10^2$ | $1.2 \cdot 10^1$ | |
| | Mass transport (mean mass) [kg/d] | $4.8 \cdot 10^2$ | $-4.0 \cdot 10^2$ | $4.3 \cdot 10^1$ | |
| PS | Item transport [items/hour] | $1.9 \cdot 10^3$ | $-1.0 \cdot 10^3$ | $4.3 \cdot 10^2$ | 0.30 |
| | Mass transport (median mass) [kg/d] | $2.7 \cdot 10^2$ | $-1.5 \cdot 10^2$ | $6.2 \cdot 10^1$ | |
| | Mass transport (mean mass) [kg/d] | $4.8 \cdot 10^2$ | $-2.6 \cdot 10^2$ | $1.1 \cdot 10^2$ | |
| PET | Item transport [items/hour] | $1.0 \cdot 10^3$ | $-2.4 \cdot 10^2$ | $3.8 \cdot 10^2$ | 0.62 |
| | Mass transport (median mass) [kg/d] | $5.0 \cdot 10^2$ | $-1.2 \cdot 10^2$ | $1.9 \cdot 10^2$ | |
| | Mass transport (mean mass) [kg/d] | $4.7 \cdot 10^2$ | $-1.1 \cdot 10^2$ | $1.8 \cdot 10^2$ | |
| $PO_{hard}$ | Item transport [items/hour] | $2.7 \cdot 10^2$ | $-3.1 \cdot 10^2$ | $-1.9 \cdot 10^1$ | $-0.07$ |
| | Mass transport (median mass) [kg/d] | $4.9 \cdot 10^1$ | $-5.6 \cdot 10^1$ | $-3.5 \cdot 10^0$ | |
| | Mass transport (mean mass) [kg/d] | $7.9 \cdot 10^1$ | $-9.0 \cdot 10^1$ | $-5.5 \cdot 10^0$ | |
| Other plastic | Item transport [items/hour] | $1.8 \cdot 10^2$ | $-2.7 \cdot 10^2$ | $-3.5 \cdot 10^1$ | $-0.16$ |
| | Mass transport (median mass) [kg/d] | $1.9 \cdot 10^1$ | $-2.6 \cdot 10^1$ | $-3.6 \cdot 10^0$ | |
| | Mass transport (mean mass) [kg/d] | $4.4 \cdot 10^1$ | $-6.1 \cdot 10^1$ | $-8.5 \cdot 10^0$ | |

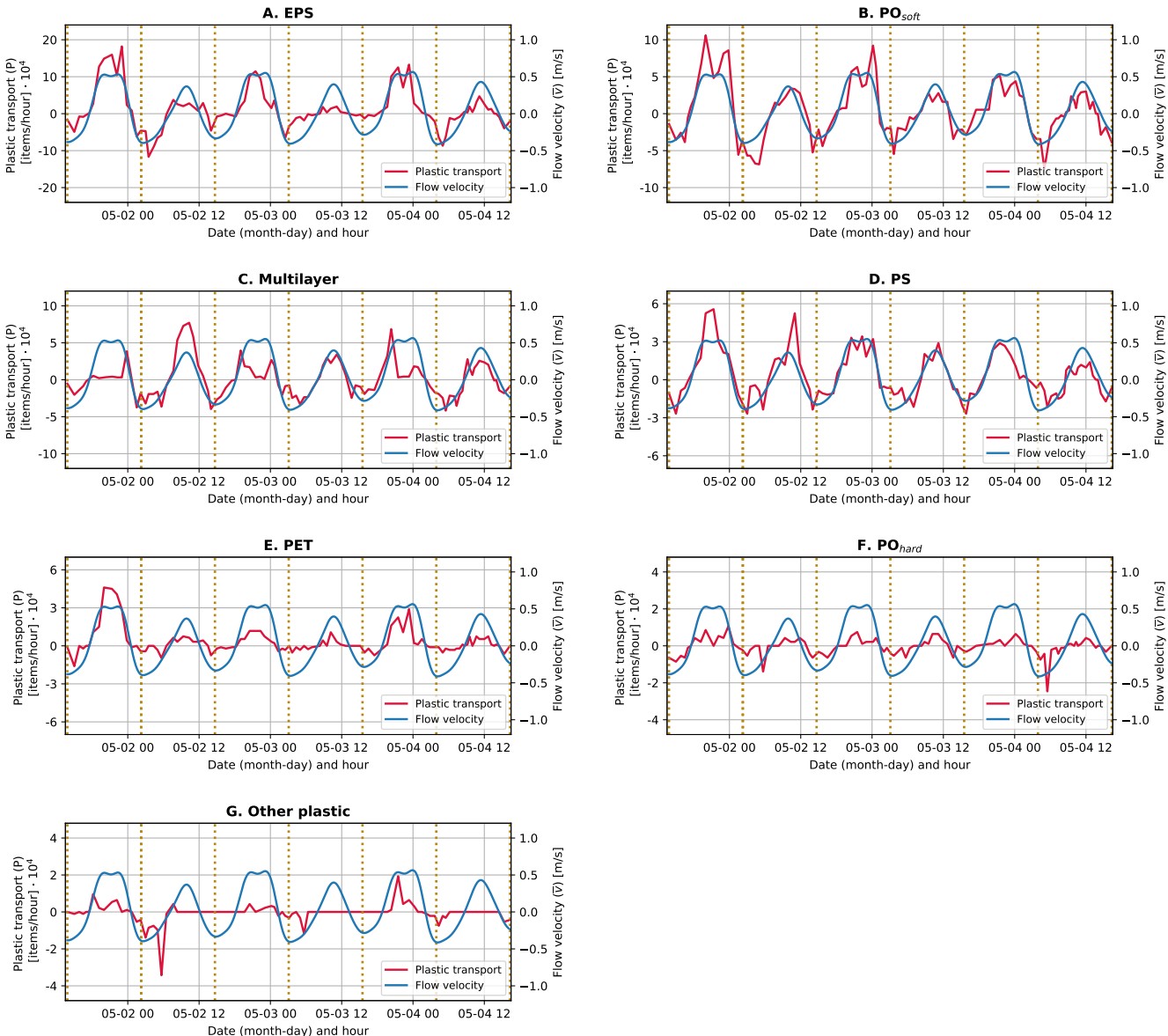

**Figure 4.** Plastic transport by item category and flow velocity over the entire measurement period (A-G). The dotted yellow lines separate each tidal cycle. The y-axis differ depending on the subplot for plastic transport, to better visualize the value distributions. Items are ranked from the most frequently found on average (EPS) to the least frequently found on average (Other plastic).

## 4    Discussion

### 4.1    Increased plastic travel distance and retention probability in tidal systems

The results of this study demonstrate that tidal dynamics strongly affect plastic transport dynamics. We found that net plastic transport corresponds to less than a third the total transport, as a result of bidirectional flows and semi-diurnal and diurnal tidal dynamics. In contrast, in non-tidal systems, without bidirectional flows, the net distance travelled by water and floating plastic equals the total traveled distance. In tidal systems however, the total distance can be much larger for the same net distance due to bidirectional flows. This is in line with previous studies that demonstrated that plastic is transported over longer total distances in estuaries compared to the freshwater reaches of rivers (Tramoy et al., 2020a).

The likelihood of plastic retention within rivers is a function of total travel distance. In tidal systems, the total travel distance per day is higher than compared to non-tidal systems, given the same net transport. Therefore, plastics have a larger probability of retention in tidal systems, for instance through deposition on riverbank and retention at hydraulic infrastructures. Various observations-based studies have highlighted the high probability of plastic retention within tidal systems (Lotcheris et al., 2023; Ledieu et al., 2022; Tramoy et al., 2020a).

In tidal systems, plastics can be retained over long periods of time, in certain cases surpassing decades as shown for the Seine river (France) (Tramoy et al., 2020b). Long retention times likely lead to high plastic concentrations, if we consider the additional inputs of plastic in and around the river. In the Saigon river, a clear seasonality in net discharge is observed. Peak net discharges (typically exceeding 200 m$^3$/s) only occur for a couple of months, usually between June and August (Camenen et al., 2021). Plastic concentrations likely only decrease significantly during these high discharge periods, due to an increase in net plastic transport and export. In this study, we only considered macroplastic (> 0.5 cm), but long macroplastic retention times would likely impact microplastic concentrations as well. Increased plastic break-down and degradation due to a prolonged presence of macroplastics in the river system probably leads to increased microplastic concentrations as well (Delorme et al., 2010; Lahens et al., 2018).

### 4.2    Neglecting tidal dynamics leads to overestimating global river plastic emission into the sea

To date, global river plastic transport and emission models do not consider tidal influence, which likely results in an overestimation of global plastic emissions into the oceans. Models that use discharge as a predictor for riverine plastic transport should be considered as export models from the non-tidal part of the river to its tidal zone, but not yet into the ocean. We found that plastic transport was strongly correlated to instantaneous discharge, which could then be used to estimate net discharge and net plastic transport. Thus, transport and emission in the tidal zone could be based on measured instantaneous discharge, instead of only using freshwater discharge estimates. Using rainfall-runoff models to estimate freshwater discharge rates entirely neglects the tidal influence on net plastic transport and emissions into the ocean. Such approaches however have been used broadly to estimate global plastic emissions (Lebreton et al., 2017; Meijer et al., 2021). Measuring discharge in tidal systems however remains very challenging and as a result, most gauging stations are located upstream of the tidal region of rivers (Gisen and Savenije, 2015; Nguyen and Nguyen, 2018). Furthermore, considering measured discharge as a more reliable

predictor of plastic transport in tidal rivers remains problematic. Establishing a fixed relation between river discharge (and other environmental drivers) and plastic transport is ultimately challenging because it cannot take into account temporal variations in plastic concentrations in the water, due to human behaviors (littering and cleaning) (Roebroek et al., 2021b).

By drawing an analogy with sediment rating curves, we can hypothesize that the rating parameters indicating availability and concentrations of plastics probably change more rapidly compared to sediment supply. The time-scales governing variability in plastic inputs into the water are likely to be shorter compared to those of sediment loads. In line with this hypothesis, Tasseron et al. (2023) observed large temporal (daily and monthly) fluctuations in plastic transport in urban waterways, a likely result of higher inputs of plastics during peak hours and seasons of outdoor human activity. The inherent difficulties in obtaining discharge estimates for tidal regions worldwide on the one hand and the limitations of using discharge as a reliable predictor of plastic transport, on the other hand, call for alternative approaches to estimating plastic emissions. Probabilistic methods that introduce a corrective factor for decreasing downstream plastic transport with decreasing distance to the river mouth could improve global transport estimates.

### 4.3 Short-term plastic transport variability driven by tidal dynamics

Our analysis has shown that plastic transport rates are highly variable over time. This temporal variability in plastic transport rates is two-fold: (i) between peak and semidiurnal-averaged net transport rates, and (ii) between the different semidiurnal-averaged net transport rates. Peak transport values ranged from $-2.1 \cdot 10^4$ to $3.7 \cdot 10^4$ items/hour over the studied period. As a consequence, field measurements that would be undertaken at the peak of either the flood and ebb flow of the tide or during a slack water phase would likely result in an overestimation or underestimation of net plastic transport. For instance, the highest mean plastic transport found during the ebb and flood phases ($2.4 \cdot 10^4$ items/hour and $-1.1 \cdot 10^4$ items/hour, respectively) are approximately one order of magnitude higher than the mean net plastic transport ($2.7 \cdot 10^3$ items/hour) for the entire measurement period. Similarly, studies on sediment transport in tidal rivers found that instantaneous peak transport values are at least one order of magnitude higher than the net (residual) sediment transport (Gatto et al., 2017). The large discrepancy between instantaneous and net plastic transport highlights the need to estimate transport rates based on longer observation periods than usually done in riverine transport studies. For example, González-Fernández et al. (2021) quantified plastic transport over 42 rivers, including five influenced by tides. Similarly, van Emmerik et al. (2022a) estimated plastic transport in Dutch rivers, encompassing 26 locations, seven of which were influenced by tides. In both studies, data collection was limited to the ebb phase, which may have led to potential overestimations of plastic transport. Furthermore, we have shown that net estimates of plastic transport vary greatly depending on whether measurements are conducted during ebb or flood dominated cycles, resulting in either positive (seaward) or negative (landward) net plastic transport, and values vary by a factor of nearly $-4$ between the highest and lowest net transport per cycle. Overall, the high variability between peak and cycle averaged net plastic transport, coupled with the variability within net plastic transport per tidal cycle highlight both the uncertainty in quantifying net plastic transport and the dependency on the temporal scale considered.

This study was the first to quantify plastic transport during full tidal cycles using a Eulerian approach. We only considered short-term tidal dynamics, namely the alternation between flood and ebb tidal phases and the diurnal cycles. Longer-term

patterns, such as the cycle in neap and spring tides, the seasonality in net discharge or peaks in freshwater discharge could all influence flow dynamics and thus significantly alter plastic transport processes. Fernandino et al. (2016) for instance observed higher floating litter densities during the spring ebb tides. This suggests that co-occurrences in hydrological conditions are also of interest when trying to understand long-term plastic transport dynamics in tidal rivers. Additional measurements of plastic transport throughout full tidal cycles of varying tidal and hydrological conditions are therefore needed for this. We therefore suggest repeating similar observations during specific conditions, such as spring/neap and high discharge/storm surge conditions. Such measurements would enable researchers to widen the range of tidal and hydrological conditions investigated in relation to plastic transport.

## 4.4 Delivery ratio of plastic is higher than water

We found that, in relative terms, plastic net transport is higher when compared with net discharge rates ($d_r$ of 16% for water flow and 20-33% for all plastic items). Two main explanations can be hypothesized for this difference in delivery ratios. The first postulates that fundamental differences exist between plastic and water transport processes. Factors not directly accounted for in this study, such as wind and different flow mobilization thresholds could impact differently water and plastics, and ultimately result in significantly higher delivery ratios for plastic compared to water. The second hypothesis relates to the site- specific dynamics. High temporary entrapment rates of plastics downstream of the measurement site could lead to lower landward transport rates compared to water, because a significant portion of items become temporarily stuck.

Hydraulic and mechanical factors, such as different motion thresholds, the influence of wind and lateral flows and sinking/resuspension mechanisms along the water columns might explain the higher delivery ratios of plastic compared to water. Our analysis has shown that during the flood phase of the tide, less plastic items were transported in the landward direction compared to water. This is somewhat surprising given that the flood phase of the tidal cycle generally corresponds to rising water levels, which could potentially mobilize items that were deposited during falling water levels (ebb phase). However, the lower flow velocities measured during the flood phase compared to the ebb phase of the tidal cycle ($-0.2$ vs 0.3 m/s) could explain that a lower share of plastic items reaches their critical threshold of motion, in contrast with water. This could be particularly relevant considering that in most rivers, including the Saigon river, plastic items are often temporarily trapped in floating vegetation, on the banks or within fluvial structures (Ledieu et al., 2022; Schreyers et al., 2021; van Emmerik et al., 2022b). Quantification of mobilization thresholds of plastics in various trapping conditions is required to further investigate this mechanism.

Besides flow velocity and discharge, other factors could influence the velocity of plastics, such as wind, waves and lateral flows (Laxague et al., 2017; van der Mheen et al., 2020). These factors could generate accelerating or decelerating effects in the propagation of plastic in the river. In addition, our study only measured floating plastic transport and therefore the influence of tidal dynamics on sub-surface plastic and the transfer of plastics between the surface and the deeper layers (sinking and re-suspension) were ignored. This is mainly due to the lack of measurement methods that are easy to deploy to quantify the distribution of plastic throughout the water column in rivers at a high temporal frequency. Tidal dynamics could also affect the vertical distribution of plastic items, due to variations in water depths and vertical mixing of fresh and salt water (Vermeiren et al., 2016). Ultimately, sinking and re-suspension mechanisms could also contribute to the higher downstream transport rate found for

plastic in comparison to water. Finally, changes in the lateral distribution of floating plastics between the ebb and flood phases could could potentially lead to higher net transport rates of plastics downstream compared to water. Specifically, if plastics are hypothesized to be more concentrated mid-stream during the ebb phase and more widely dispersed over the river width during the flood phase, this could increase the likelihood of entrapment along the riverbanks during upstream transport. This could contribute to higher net plastic transport in the downstream direction. However, this specific scenario was not supported by our findings (Schreyers et al., 2023).

Another hypothesis pertains to the local characteristics of our case-study area. High rates of plastic entrapment/deposition downstream of the measurement site, compared to upstream could explain the relative lower landward transport rates compared to water. High concentrations of items were often found downstream of the measurement site, due to the presence of docks, piers and jetties which temporarily trap items (Lotcheris et al., 2023). Similar trapping elements were not found directly upstream of the measurement site. Other factors such as the vegetation, wood jams and meandering might also influence plastic accumulation rates on riverbanks, as already evidenced by recent research (Ledieu et al., 2022; Liro et al., 2020)). The two hypotheses presented for higher delivery ratios of plastics compared to water could be tested using Lagrangian approaches, in combination with high frequency hydrometeorological measurements throughout tidal cycles. Lagrangian studies on plastic transport could provide insights into the (re)mobilization and entrapment thresholds in relation to flow and other hydrometeorological factors such as wind. To the best of our knowledge, no Lagragian-based approaches have so far quantified thresholds for the mobilization and stopping of mobile plastics. In addition, Lagrangian approaches are also useful in mapping entrapment/accumulation zones along a river course (Ledieu et al., 2022).

## 4.5 Plastic transport processes are affected by the geometry, size and buoyancy of items

Our results show that different plastic categories have highly variable net transport rates, depending on items type characteristics, such as size, rigidity and buoyancy. Large and highly buoyant plastics were found to have higher downstream net transport rates than smaller and more neutrally buoyant items. PET items (mainly bottles) were the largest category of plastics by size (average size: 20 cm vs 11 cm for all item categories) and had the highest delivery ratio found (62%). Highly buoyant items such as EPS and PS items (food containers, isolation foam, cups and plates), with densities comprised between 0.016 and 0.640 g cm$^3$ for EPS and between 1.01 and 1.04 g cm$^3$ for PS (van Emmerik and Schwarz, 2020) were found to have high downstream net transport rates (38% and 30%, respectively). Such items are also more prone to wind influence (Jackson, 1998; Schwarz et al., 2019). This could cause both deposition effects on the sides of the river or on the riverbanks, or longer travel distances over the same duration than other items, depending on the wind direction and magnitude. Ryan (2021); López et al. (2020) found that highly buoyant plastics travel longer distances between the coast and the marine environment. In addition, because of their high buoyancy, these items do not sink easily in the water column (Schwarz et al., 2019). All these factors could explain the higher net export ratios found for highly buoyant plastics. In comparison, more neutrally buoyant and soft items such as PO$_{soft}$ (bags and foils) and Multilayer items (food wrapping) (van Emmerik et al., 2019) had lower net transport rates than average (between 10% and 16% vs 25% for all plastics). Because of their lower buoyancy, such items are more prone to vertical mixing and the influence of changes in turbulence and density fronts, such as salt concentrations (Acha et al., 2003; Ballent

et al., 2012). This is particularly relevant for tidal rivers and estuaries, due to changes in the relative balance between fresh and salt water and higher turbulence resulting from the changes in density distribution, compared to the freshwater reaches of a river.

These findings confirm that, similarly to sediment, plastic transport processes should be studied in relation to items characteristics, instead of considering plastics as a single uniform type of material (Kooi et al., 2018; Schwarz et al., 2019). The wide range of sizes, geometry, densities, buoyancy and masses of plastics strongly impacts their transport dynamics (both vertically and horizontally), as already pointed out by several studies (Ryan, 2021; Waldschläger and Schüttrumpf, 2019; Kuizenga et al., 2021). Comparably, sediment grain size distribution and density strongly influence settling and advection velocities of particles in the water. Recent sediment transport models that incorporated a broader distribution range of grain sizes and densities led to improved estimates of suspended sediment loads compared to models which used more uniform distributions (Lepesqueur et al., 2019).

## 5 Conclusions

For the first time, we quantified net plastic transport over full tidal cycles in a tidal river using a Eulerian approach. Over this time-period, we conducted sub-hourly measurements of flow velocity, water depth and plastic transport. Time-series of flow velocity and discharge estimates were extrapolated by fitting the tidal constituents of our observations, for which we found that the semi-diurnal and diurnal components were the most significant. We introduced a simple Eulerian approach, which expresses net transport by establishing a balance between the flood (landward) and ebb (seaward) water flows and plastic transport. This approach could easily be transferred to other river systems as it requires limited and affordable equipment.

Four main findings on plastic transport in tidal regions are highlighted from our study. First, net plastic transport is lower compared to total transport, due to estuarine dynamics. In our case-study, we found that net transport amounted to only 20-33% of the total plastic transport. This likely leads to longer travel distances of plastics in tidal systems compared to non-tidal ones, facilitating plastic retention along the river course. Secondly, estimates of river plastic transport show high short-term variability due to tidal dynamics. Diurnal inequality in the tides causes an alternation between cycles with positive net transport (seaward plastic transport) and cycles with negative net transport (i.e.: landward plastic transport). We also found that peak and semidiurnal averaged net transport rates varied by as much as one order of magnitude. Thirdly, net plastic transport shows higher net downstream transport compared to water. We found that net water discharge amounted to 16% of the total river discharge, whereas net plastic transport corresponds to 20-33% of the total plastic transport. This suggests that either plastic travel longer distances than water, possibly due to the influence of other environmental drivers such as wind, or that plastics are get often trapped downstream from the measurement site, limiting their transport upstream during the flood tidal phase. Lastly, plastics are not uniformly affected by tidal dynamics. Larger and highly buoyant items, such as plastic foams and polystyrene have larger net transport ratios compared to neutrally buoyant and more flexible items, such as bags, foils and food packaging.

In this paper, we show that tidal dynamics play a crucial role in total and net plastic transport in tidal rivers. Bidirectional flows resulting from the semi-diurnal tidal component lead to a large discrepancy between net and total plastic transport rates. With each river that flows into the ocean being affected by tidal dynamics, such effects cannot be neglected anymore in studies

that quantify (global) plastic emissions into the ocean. Efforts to both conceptualize and integrate tidal dynamics in river plastic

transport and emissions models are therefore required.

## Appendix A:  Relationship between river discharge and flow velocity

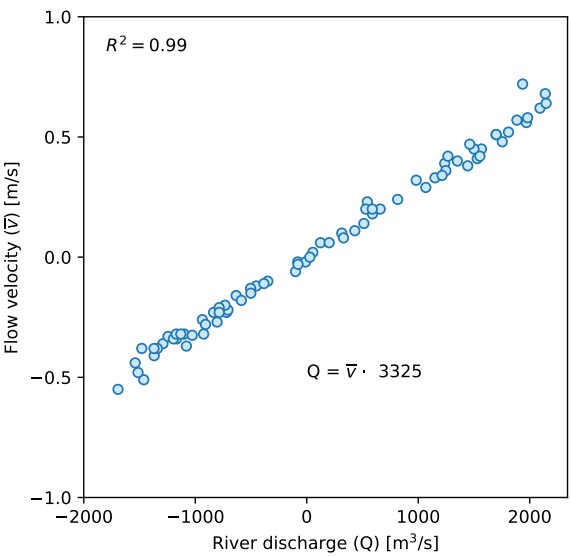

**Figure A1.** Relationship between river discharge and flow velocity. p-value <0.01.

## Appendix B:  Tidal constituent analysis

We found M2 (principal lunar semi-diurnal) and K1 (lunar diurnal) to be the dominating tidal constituents over our flow velocity time-series (Table B1). However, the distortions of the sinusoidal symmetry (Fig.3) could be attributed to shallow water override components (M4 and M6), which were also found to be significant and/or the interactions between the M2 and K1 components (Hoitink et al., 2003; Gatto et al., 2017).

**Table B1.** Tidal constituent coefficients (amplitude and frequency) and signal-to-noise ratio.

| Tidal constituent | Symbol | Amplitude [m/s] | Frequency [cycles/hour] | Signal-to-Noise ratio [-] |
|---|---|---|---|---|
| Principal lunar semi-diurnal | M2 | 0.43 | 0.08 | 972 |
| Lunar diurnal | K1 | 0.14 | 0.04 | 8987 |
| Fifth diurnal | 2MK5 | 0.05 | 0.20 | 27 |
| Shallow water overtide of principal lunar | M4 | 0.04 | 0.16 | 7 |
| Shallow water overtide of principal lunar | M6 | 0.03 | 0.24 | 96 |
| Seventh diurnal | 3MK7 | 0.02 | 0.28 | 4 |
| Lunar terdiurnal | M3 | 0.01 | 0.12 | 1 |
| Shallow water eight diurnal | M8 | $< 0.01$ | 0.32 | $< 1$ |

## Appendix C:  Relationship between river discharge and plastic transport

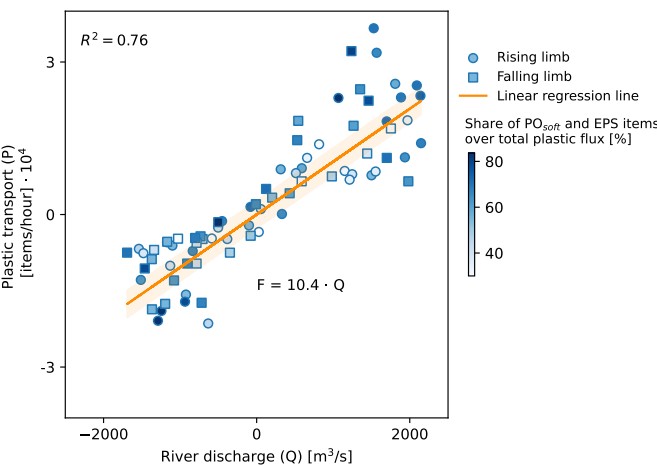

**Figure C1.** Relationship between plastic transport and river discharge. p-value < 0.01.

**Figure C2.** Relationship between plastic transport and river discharge by plastic types (A-G). All p-values were found to be below <0.01.

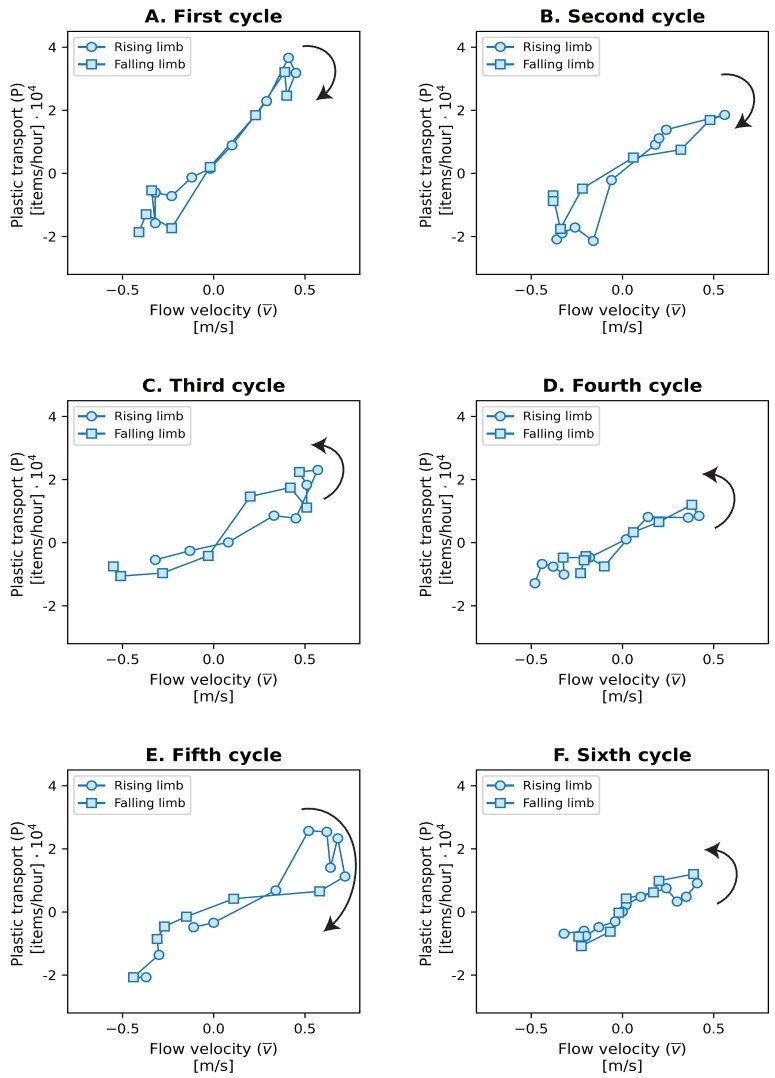

**Figure C3.** Relationship between plastic transport and flow velocity per tidal cycle (A-F). The arrows indicate the direction of the hysteresis (clockwise or anticlockwise) between rising and falling limbs of the tidal cycle.

## Appendix D: Plastic transport, flow velocity, discharge, and associated metrics: extended data

### Alternative calculation methods to estimate delivery ratios

We propose two alternative methods for calculating delivery ratios ($d_r$). In Eq. 6, the denominator corresponds to the total transported amounts/volumes or distances ($V_t$) for the variable $f$ considered. A potential issue with using $V_t$ as our denominator in Eq.6 could be that $V_e$ and $V_f$ can be seen as not being independent of each other, because part of $V_e$ is likely to be circulated in $V_f$ and vice-versa. To address this, we introduce an alternative delivery ratio, $d_{r,2}$:

$$d_{r,2} = \frac{\overbrace{V_e + V_f}^{\substack{\text{Net transported amounts/}\\\text{volumes/distances } (V_n)}}}{\underbrace{(V_e - V_f) \cdot \frac{1}{2}}_{\substack{\text{Mean transported amounts/}\\\text{volumes/distances } (V_m)}}} \tag{D1}$$

Here, we consider the mean transported amounts/volumes/distances as the denominator. However, the delivery ratios can exceed unity for plastics (see Table D2). We aimed at constraining the delivery values between $-1$ and $1$, to ease the interpretation, with a value of zero indicating no net transport over the tidal cycle and a value of 1 or $-1$ indicates that the total volume of plastic has been transported downstream or upstream, respectively. We therefore suggest the following equation as another alternative to calculate delivery ratio, $d_{r,3}$:

$$d_{r,3} = \frac{\overbrace{V_e + V_f}^{\substack{\text{Net transported amounts/}\\\text{volumes/distances } (V_n)}}}{\underbrace{max(V_e, |V_f|)}_{\substack{\text{Maximum transported amounts/}\\\text{volumes/distances between ebb and flood}}}} \tag{D2}$$

By taking the maximum value between the plastic volume during ebb and the absolute amounts/volumes/distances during flood, we constraint our delivery values between -1 and 1, because the denominator cannot be smaller than the numerator in such a case.

     We report results based on $d_{r,1}$, $d_{r,2}$, and $d_{r,3}$ in Tables D1 and D2. Regardless of the selected delivery ratio calculation
method, our main conclusions are supported. Plastics delivery is higher than that of water for all $d_{r,1}$, $d_{r,2}$, and $d_{r,3}$ values (Tables D1 and D2). Ebb-dominated cycles have positive delivery values (net downstream transport), whereas flood-dominated cycles are characterized by negative delivery values (net upstream transport) (Table D2). Our finding that net plastic transport is limited by tidal dynamics remains correct, as delivery values over the entire monitored period were all found to be below unity (Table D1). The extent of the reduction in transport is, however, variable depending on the calculation method chosen. As we
want to constrain delivery values to be between $+1/-1$ and 0, we consider $d_{r,2}$ not to be a suitable option. We reported $d_{r,1}$ values in the main manuscript.

**Table D1.** Summary of plastic transport, flow velocity, discharge, and associated metrics during ebb and flood phases. $d_{r,1}$, $d_{r,2}$ and $d_{r,3}$ correspond to the three calculation methods for the delivery ratio, as presented in Appendix D. In the main manuscript we only report $d_{r,1}$, for brevity.

| | $f_e$ | $f_f$ | $f_n$ |
|---|---|---|---|
| Mass transport (median mass) [kg/d] | $1.4 \cdot 10^3$ | $-7.2 \cdot 10^2$ | $3.5 \cdot 10^2$ |
| Mass transport (mean mass) [kg/d] | $2.8 \cdot 10^3$ | $-1.9 \cdot 10^3$ | $4.7 \cdot 10^2$ |
| Item transport [items/hour] | $1.3 \cdot 10^4$ | $-8.0 \cdot 10^3$ | $2.7 \cdot 10^3$ |
| River discharge [m³/s] | $1.1 \cdot 10^3$ | $-8.1 \cdot 10^2$ | $1.6 \cdot 10^2$ |
| Flow velocity [m/s] | $0.3$ | $-0.2$ | $> 0.0$ |

| | $V_e$ | $V_F$ | $V_n$ | $d_{r,1}$ | $d_{r,2}$ | $d_{r,3}$ |
|---|---|---|---|---|---|---|
| | | | | [-] | [-] | [-] |
| Mass (median mass per item) [kg] | $2.2 \cdot 10^3$ | $-1.1 \cdot 10^3$ | $1.1 \cdot 10^3$ | 0.33 | 0.51 | 0.36 |
| Mass (mean mass per item) [kg] | $4.4 \cdot 10^3$ | $-2.9 \cdot 10^3$ | $1.5 \cdot 10^3$ | 0.20 | 0.65 | 0.46 |
| Number of items [items] | $5.0 \cdot 10^5$ | $-3.0 \cdot 10^5$ | $2.0 \cdot 10^5$ | 0.25 | 0.40 | 0.28 |
| Water volume [m³] | $1.5 \cdot 10^8$ | $-1.1 \cdot 10^8$ | $4.3 \cdot 10^7$ | 0.16 | 0.32 | 0.25 |
| Distance [m] | $4.5 \cdot 10^4$ | $-3.2 \cdot 10^4$ | $1.3 \cdot 10^4$ | 0.16 | 0.32 | 0.25 |

**Table D2.** Plastic transport, flow velocity, discharge, and associated metrics during ebb and flood phases, per tidal cycle. $d_{r,1}$, $d_{r,2}$ and $d_{r,3}$ correspond to the three calculation methods for the delivery ratio, as presented in Appendix D. In the main manuscript we only report $d_{r,1}$, for brevity.

| Cycles | Variables | Plastic transport | | | Flow velocity / distance | River discharge / water volume |
| | | Item transport / amount [items/hour] or [items] | Mass transport / mass (median mass per item) [kg/d] or [kg] | Mass transport / mass (mean mass per item) [kg/d] or [kg] | [m/s] or [m] | [m³/s] or [m³] |
|---|---|---|---|---|---|---|
| 1 | $f_e$ | $2.4 \cdot 10^4$ | $2.8 \cdot 10^3$ | $5.6 \cdot 10^3$ | $0.4$ | $1.3 \cdot 10^3$ |
| | $f_f$ | $-8.3 \cdot 10^3$ | $-8.1 \cdot 10^2$ | $-2.0 \cdot 10^3$ | $-0.3$ | $-8.8 \cdot 10^2$ |
| | $f_n$ | $9.7 \cdot 10^3$ | $1.2 \cdot 10^3$ | $2.3 \cdot 10^3$ | $0.1$ | $3.7 \cdot 10^2$ |
| | $V_e$ | $1.7 \cdot 10^5$ | $8.6 \cdot 10^2$ | $1.6 \cdot 10^3$ | $1.0 \cdot 10^4$ | $3.4 \cdot 10^7$ |
| | $V_f$ | $-4.5 \cdot 10^4$ | $-1.8 \cdot 10^2$ | $-4.5 \cdot 10^2$ | $-5.2 \cdot 10^3$ | $-1.7 \cdot 10^7$ |
| | $V_n$ | $1.2 \cdot 10^5$ | $6.3 \cdot 10^2$ | $1.2 \cdot 10^3$ | $4.9 \cdot 10^3$ | $1.6 \cdot 10^7$ |
| | $d_{r,1}$ [-] | $0.57$ | $0.63$ | $0.57$ | $0.32$ | $0.32$ |
| | $d_{r,2}$ [-] | $1.15$ | $1.27$ | $1.13$ | $0.65$ | $0.65$ |
| | $d_{r,3}$ [-] | $0.73$ | $0.78$ | $0.72$ | $0.49$ | $0.49$ |
| 2 | $f_e$ | $1.3 \cdot 10^4$ | $1.2 \cdot 10^3$ | $2.5 \cdot 10^3$ | $0.2$ | $7.3 \cdot 10^2$ |
| | $f_f$ | $-1.1 \cdot 10^4$ | $-9.1 \cdot 10^2$ | $-2.6 \cdot 10^3$ | $-0.3$ | $-8.1 \cdot 10^2$ |
| | $f_n$ | $-1.8 \cdot 10^3$ | $-5.0 \cdot 10^1$ | $-5.6 \cdot 10^2$ | $-0.1$ | $-1.9 \cdot 10^2$ |
| | $V_e$ | $6.2 \cdot 10^4$ | $2.6 \cdot 10^2$ | $5.1 \cdot 10^2$ | $3.9 \cdot 10^3$ | $1.3 \cdot 10^7$ |
| | $V_f$ | $-8.4 \cdot 10^4$ | $-2.8 \cdot 10^2$ | $-8.0 \cdot 10^2$ | $-6.5 \cdot 10^3$ | $-2.2 \cdot 10^7$ |
| | $V_n$ | $-2.2 \cdot 10^4$ | $-2.6 \cdot 10^1$ | $-2.9 \cdot 10^2$ | $-2.6 \cdot 10^3$ | $-8.7 \cdot 10^6$ |
| | $d_{r,1}$ [-] | $-0.15$ | $-0.05$ | $-0.22$ | $-0.25$ | $-0.25$ |
| | $d_{r,2}$ [-] | $-0.30$ | $-0.10$ | $-0.44$ | $-0.50$ | $-0.50$ |
| | $d_{r,3}$ [-] | $-0.26$ | $-0.09$ | $-0.36$ | $-0.40$ | $-0.40$ |
| 3 | $f_e$ | $1.4 \cdot 10^4$ | $1.3 \cdot 10^3$ | $2.7 \cdot 10^3$ | $0.4$ | $1.4 \cdot 10^3$ |
| | $f_f$ | $-6.4 \cdot 10^3$ | $-5.9 \cdot 10^2$ | $-1.5 \cdot 10^3$ | $-0.2$ | $-8.1 \cdot 10^2$ |
| | $f_n$ | $5.3 \cdot 10^3$ | $5.1 \cdot 10^2$ | $8.6 \cdot 10^2$ | $0.1$ | $4.3 \cdot 10^2$ |
| | $V_e$ | $1.0 \cdot 10^5$ | $3.9 \cdot 10^2$ | $7.9 \cdot 10^2$ | $1.0 \cdot 10^4$ | $3.5 \cdot 10^7$ |
| | $V_f$ | $-3.4 \cdot 10^4$ | $-1.3 \cdot 10^2$ | $-3.4 \cdot 10^2$ | $-4.7 \cdot 10^3$ | $-1.5 \cdot 10^7$ |
| | $V_n$ | $6.6 \cdot 10^4$ | $2.6 \cdot 10^2$ | $4.5 \cdot 10^2$ | $5.8 \cdot 10^3$ | $1.9 \cdot 10^7$ |
| | $d_{r,1}$ [-] | $0.49$ | $0.50$ | $0.40$ | $0.39$ | $0.39$ |
| | $d_{r,2}$ [-] | $0.98$ | $1.00$ | $0.79$ | $0.77$ | $0.77$ |
| | $d_{r,3}$ [-] | $0.66$ | $0.67$ | $0.57$ | $0.56$ | $0.56$ |
| 4 | $f_e$ | $6.3 \cdot 10^3$ | $6.8 \cdot 10^2$ | $1.4 \cdot 10^3$ | $0.2$ | $7.8 \cdot 10^2$ |
| | $f_f$ | $-6.5 \cdot 10^3$ | $-6.1 \cdot 10^2$ | $-1.6 \cdot 10^3$ | $-0.2$ | $-8.0 \cdot 10^2$ |
| | $f_n$ | $-9.7 \cdot 10^2$ | $-5.6 \cdot 10^1$ | $-3.3 \cdot 10^2$ | $<0.0$ | $-1.2 \cdot 10^2$ |
| | $V_e$ | $3.4 \cdot 10^4$ | $1.5 \cdot 10^2$ | $3.0 \cdot 10^2$ | $4.5 \cdot 10^3$ | $1.5 \cdot 10^7$ |
| | $V_f$ | $-4.6 \cdot 10^4$ | $-1.8 \cdot 10^2$ | $-4.7 \cdot 10^2$ | $-6.2 \cdot 10^3$ | $-2.0 \cdot 10^7$ |
| | $V_n$ | $-1.2 \cdot 10^4$ | $-2.9 \cdot 10^1$ | $-1.7 \cdot 10^2$ | $-1.6 \cdot 10^3$ | $-5.4 \cdot 10^6$ |
| | $d_{r,1}$ [-] | $-0.15$ | $-0.09$ | $-0.22$ | $-0.15$ | $-0.15$ |
| | $d_{r,2}$ [-] | $-0.30$ | $-0.18$ | $-0.44$ | $-0.31$ | $-0.31$ |
| | $d_{r,3}$ [-] | $-0.26$ | $-0.16$ | $-0.36$ | $-0.27$ | $-0.27$ |
| 5 | $f_e$ | $1.4 \cdot 10^4$ | $1.5 \cdot 10^3$ | $3.0 \cdot 10^3$ | $0.4$ | $1.4 \cdot 10^3$ |
| | $f_f$ | $-4.2 \cdot 10^3$ | $-4.9 \cdot 10^2$ | $-1.1 \cdot 10^3$ | $-0.2$ | $-7.5 \cdot 10^2$ |
| | $f_n$ | $6.7 \cdot 10^3$ | $6.7 \cdot 10^2$ | $1.3 \cdot 10^3$ | $0.2$ | $4.9 \cdot 10^2$ |
| | $V_e$ | $1.0 \cdot 10^5$ | $4.5 \cdot 10^2$ | $9.0 \cdot 10^2$ | $1.1 \cdot 10^4$ | $3.6 \cdot 10^7$ |
| | $V_f$ | $-2.2 \cdot 10^4$ | $-1.1 \cdot 10^2$ | $-2.3 \cdot 10^2$ | $-4.2 \cdot 10^3$ | $-1.4 \cdot 10^7$ |
| | $V_n$ | $8.3 \cdot 10^4$ | $3.4 \cdot 10^2$ | $6.6 \cdot 10^2$ | $6.5 \cdot 10^3$ | $2.2 \cdot 10^7$ |
| | $d_{r,1}$ [-] | $0.66$ | $0.62$ | $0.59$ | $0.44$ | $0.44$ |
| | $d_{r,2}$ [-] | $1.32$ | $1.24$ | $1.17$ | $0.87$ | $0.87$ |
| | $d_{r,3}$ [-] | $0.79$ | $0.76$ | $0.74$ | $0.61$ | $0.61$ |
| 6 | $f_e$ | $5.6 \cdot 10^3$ | $5.7 \cdot 10^2$ | $1.1 \cdot 10^3$ | $0.3$ | $8.5 \cdot 10^2$ |
| | $f_f$ | $-9.8 \cdot 10^3$ | $-8.6 \cdot 10^2$ | $-2.3 \cdot 10^3$ | $-0.2$ | $-8.0 \cdot 10^2$ |
| | $f_n$ | $-2.7 \cdot 10^3$ | $-2.0 \cdot 10^2$ | $-7.0 \cdot 10^2$ | $< 0.0$ | $-3.4 \cdot 10^1$ |
| | $V_e$ | $3.2 \cdot 10^4$ | $1.4 \cdot 10^2$ | $2.7 \cdot 10^2$ | $5.3 \cdot 10^3$ | $1.8 \cdot 10^7$ |
| | $V_f$ | $-6.5 \cdot 10^4$ | $-2.4 \cdot 10^2$ | $-6.4 \cdot 10^2$ | $-5.7 \cdot 10^3$ | $-1.9 \cdot 10^7$ |
| | $V_n$ | $-3.3 \cdot 10^4$ | $-1.0 \cdot 10^2$ | $-3.6 \cdot 10^2$ | $-4.5 \cdot 10^2$ | $-1.5 \cdot 10^6$ |
| | $d_{r,1}$ [-] | $-0.34$ | $-0.27$ | $-0.40$ | $-0.04$ | $-0.04$ |
| | $d_{r,2}$ [-] | $-0.68$ | $-0.54$ | $-0.80$ | $-0.08$ | $-0.08$ |
| | $d_{r,3}$ [-] | $-0.51$ | $-0.43$ | $-0.57$ | $-0.08$ | $-0.08$ |

**Table D3.** Mean and median mass per item. The mass statistics were taken from van Emmerik et al. (2019). The reported values for Multilayer and Other plastic correspond to the mean and median for all items, since mass was not measured for a sufficient number of items for these two categories.

| Plastic type | EPS | $PO_{soft}$ | Multilayer | PS | PET | $PO_{hard}$ | Other plastic |
|---|---|---|---|---|---|---|---|
| Mean mass per item [g] | 7.0 | 10.6 | 10.6 | 10.7 | 20.0 | 12.3 | 10.1 |
| Median mass per item [g] | 1.9 | 2.9 | 2.9 | 6.0 | 20.8 | 7.7 | 4.3 |

*Author contributions.* Conceptualization: LS and TvE; Investigation - Data collection: LS, TvE, KB, KvLT; Formal analysis: LS; Visualization: LS; Data curation: LS; Writing-original draft: LS; Writing-reviewing and editing: all authors; Supervision: MvdP; Project administration: TvE and LS; Funding acquisition: TvE and LS.

*Competing interests.* The authors declare having no competing interests

*Acknowledgements.* Thanks to Quynh Nhu, Thanh Phng, My Linh, Do Tien, Minh Son, Quoc Dung, Hoang Linh, Tran Danh, Diem Quynh, Anh Thu, Phuong Uyen and Thu Trúc for their help in data collection. Thank you to Henk Jongbloed, Joris Beemster, Nick Wallerstein and Ton Hoitink for their meaningful inputs on the manuscript. The work of LS was supported by the Discovery Element of the European Space Agency's Basic Activities (ESA contract no. 4000132682/20/NL/GLC). The work of TvE was supported by the Veni Research Program, the 525 River Plastic Monitoring Project with project number 18211, which was (partly) financed by the Dutch Research Council (NWO).

## Data availability

The data underlying this manuscript will be made available upon publication at: 10.4121/21732818. For reviewers, the dataset is already accessible at: https://figshare.com/s/84ba02a91b8d1864ab48.

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
