# Peer review of "River plastic transport affected by tidal dynamics"

_EGUsphere, 2022_

## Author Comment (AC1)

**Review of egusphere-2022-1495**

I consider this a very relevant paper. It is well written and reports on extensive field observations in the Saigon tidal river, which I consider very valuable. Also the conclusion that the plastic travels faster than the fresh water component of the tidal flow is relevant. But the calculation of the tidal volumes and efficiencies is, unfortunately, doubtful because of erroneous or unclear mathematics. So, until this unclarity is solved, we cannot judge if this conclusion is correct.

**Thank you very much for your constructive feedback on the manuscript. We appreciate your inputs which will lead to the improvement of the manuscript. We agree that the equations used were at times unclear. Here, we clarify our initial delivery ratio equation, and will do so as well in a revised version of the manuscript, which we will submit once we receive additional reviews. Considering your feedback on our calculations, we suggest three different equations (eq. 1, 4 and 6) to express the delivery ratio and present in this rebuttal the main results using these equations (see Tables R1 and R2 in this file). Independently from the equations used, our main findings that plastic net exports are limited due to tidal dynamics and that plastic travels faster than water remain valid (Table R1). Unless fundamental flaws are highlighted regarding the calculation of delivery values using the total volume (the sum of flood and ebb volumes), we will keep our initial definition of delivery ratio, now formulated in equation 1 of this rebuttal.**

In line 155, the authors mention that ebb flow is positive and flood flow is negative. Fair enough. But in that case, the integral of the tidal volumes, V_ebb and V_flood, should also be positive and negative.

**Thank you for pointing that out. Indeed, the volumes during ebb and flood phases ($V_E$ and $V_F$) are respectively positive and negative. Some of our previous equations used absolute values, some did not. We have now harmonized our equations and do not use absolute values anymore. We will make sure that we are consistent on this aspect in a revised version of the manuscript.**

This makes the calculation of the delivery ratio unclear. This calculation should then be:

d_r= Delta(V)/V_tidal or

d_r= (V_ebb+V_flood)/(V_tidal)

V_tidal should be the average of the ebb and flood volumes: (Vebb-Vflood)/2.

If, instead the absolute volumes of the ebb and flood flow are taken, then the equation becomes:

d_r= 2*(V_ebb-V_flood)/(V_ebb+V_flood)

This would be:

In the case of plastic transport: the delivery of plastic to downstream in relation to the tidal transport of plastic.

In the case of river flow: the delivery of fresh water over the tidal volume or (Q_fresh*T)/(Tidal volume), which is approximately equal to the ratio between the fresh water velocity and the average ebb or flood tidal velocity.

One would expect these two delivery ratios to be the same, but interestingly they are not.

Now three things are wrong in Eq.(6)

First, the negative sign in the numerator and the positive sign in the denominator, unless absolute values of the volumes are implied, but then Eq.(9) would be wrong; it should then have a minus in the numerator.

**Indeed, we used absolute values in equation 6, and signed values in equation 9. We will revise this. We choose to not report absolute values.**

Second the factor 2. There should be a factor 2 in the numerator of Eq.(6)

**Regarding the factor 2 in equation 6, we think this depends on what we consider to be our volume of reference. Our goal with introducing this equation was to provide a metric to assess the reduction in transport/flow due to tidal dynamics. Figure R1 provides an overview of the terms used in the equations presented below.**

[Figure]

**Figure R1.** Schematic graph on variables for delivery ratio ($d$) calculation. $V_E$: volume during ebb phase, $V_F$: volume during flood phase, $V_N$: net volume, $V_T$: total volume, $V_M$: mean tidal volume.

**Our initial idea was to calculate the delivery ratio as follows:**

$$d_1 = \frac{V_N}{V_T}$$

(1)

$V_N$ is here the net volume and $V_T$ the total volume passing through a cross-section during a tidal cycle (Figure R1). Since $V_E$ is positive and $V_F$ is negative, we calculate $V_N$ using the following equation:

$$V_N = V_E + V_F$$

(2)

Similarly, we calculate the $V_T$ by using:

$$V_T = V_E - V_F$$

(3)

Apart from the initial unclarity in the delivery ratio equation, we hypothesized that a potential issue with using $V_T$ as our denominator in eq.1 could be that $V_E$ and $V_F$ can be seen as not being independent of each other, because part of $V_E$ is likely to be circulated in $V_F$ and vice-versa. We also considered your suggestion, to use the mean tidal volume ($V_M$) instead:

$$d_2 = \frac{V_N}{V_M}$$

(4)

$$V_M = \frac{V_T}{2}$$

(5)

An issue however arises when using the mean tidal volume as the denominator: delivery values can exceed unity in the case of plastics. We would like to constrain the delivery values to be comprised between -1 and 1. A value of zero would indicate no net transport over the tidal cycle and a value of 1 or -1 indicates that the total volume of plastic has been transported downstream or upstream, respectively.

We therefore suggest the following equation as another option:

$$d_3 = \frac{V_N}{\max{(V_E, |V_F|)}}$$

(6)

By taking the maximum value between the plastic volume during ebb and the absolute plastic volume during flood, we constrain our delivery values between -1 and 1, because the denominator cannot be smaller than the numerator in such case.

Please note that the calculation of $d_3$ over the entire monitored period (values reported in Table R1) required additional choices regarding how $V_E$ and $V_F$ are calculated. This could be either done for each tidal cycle individually, or for the entire period of interest. We used the following equations to calculate $d_3$ over the entire monitoring period:

$$d_3 = \frac{\sum_{i=1}^{6} V_N}{\sum_{i=1}^{6} \max{(V_E, |V_F|)}}$$

(7)

In this way, we use the sum for the maximum value between $V_E$ and $V_F$ for each tidal cycle, instead of using the maximum between $V_E$ and $V_F$ calculated over the entire period. We considered this a suitable choice, given that in our case, there is an alternation between flood and ebb-dominant cycles. Thus using the sum of maximum values between $V_E$ and $V_F$ for each tidal cycle seemed a more accurate way for estimating the maximum volume between $V_E$ and $V_F$.

Figure R1 helps to explain the three different equations proposed, which we needed for our conceptualization and might help future reviewers.

We summarize here the results in Table R1 and R2, with the three delivery values calculated as presented above. We have found some errors in the division between ebb and flood phases in our data, therefore our net transport values ($P_N$) and $d_1$ have also slightly changed compared to those reported in the submitted manuscript. But the changes are minimal. As you will notice, certain values of $d_2$ are above unity. This was found to be the case for the delivery value of item transport during the first tidal cycle, as well as the fifth tidal cycle (Table R2).

**Table R1.** Summary statistics for plastic, flow velocity and discharge. $P_E$: transport during ebb phase. $P_F$: transport during flood phase, $P_N$: net transport, $V_T$: total volume, $V_M$: mean tidal volume, $V_E$: volume during ebb phase. $V_F$: volume during flood phase. $V_N$: net volume.

| | $P_E$ | $P_F$ | $P_N$ |
|---|---|---|---|
| Mass transport (median mass) [kg/day] | $1.6 \cdot 10^3$ | $-8.6 \cdot 10^2$ | $3.9 \cdot 10^2$ |
| Mass transport (mean mass) [kg/day] | $3.3 \cdot 10^3$ | $-2.1 \cdot 10^3$ | $6.0 \cdot 10^2$ |
| Item transport [items/hour] | $1.5 \cdot 10^4$ | $-9.0 \cdot 10^3$ | $3.0 \cdot 10^3$ |
| River discharge [m³/s] | $1.1 \cdot 10^3$ | $-8.3 \cdot 10^2$ | $1.6 \cdot 10^2$ |
| Flow velocity [m/s] | 0.3 | -0.2 | >0.0 |

| | $V_E$ | $V_F$ | $V_N$ |
|---|---|---|---|
| Mass (median mass per item) [kg] | $2.6 \cdot 10^3$ | $-1.3 \cdot 10^3$ | $1.2 \cdot 10^3$ |
| Mass (mean mass per item) [kg] | $5.1 \cdot 10^3$ | $-3.3 \cdot 10^3$ | $1.9 \cdot 10^3$ |
| Total number of items [items] | $5.6 \cdot 10^5$ | $-3.3 \cdot 10^5$ | $2.2 \cdot 10^5$ |
| Water volume [m³] | $1.5 \cdot 10^8$ | $-1.1 \cdot 10^8$ | $4.3 \cdot 10^7$ |
| Tidal excursion length [m] | $4.5 \cdot 10^4$ | $-3.2 \cdot 10^4$ | $1.3 \cdot 10^4$ |

| | $d_1$ | $d_2$ | $d_3$ |
|---|---|---|---|
| Mass transport (median mass) | 0.31 | 0.62 | 0.45 |
| Mass transport (mean mass) | 0.22 | 0.45 | 0.32 |
| Item transport | 0.25 | 0.50 | 0.35 |
| River discharge / Flow velocity | 0.16 | 0.32 | 0.25 |

**Table R2.** Net plastic transport, flow velocity and river discharge and associated delivery ratios by tidal cycle. Each tidal cycle lasts 12 hours and 25 minutes. $P_N$: net transport.

| | Cycle | 1 | 2 | 3 | 4 | 5 | 6 |
|---|---|---|---|---|---|---|---|
| $P_N$ | Item transport [items/hour] | $1.1 \cdot 10^4$ | $-2.0 \cdot 10^3$ | $5.8 \cdot 10^3$ | $-9.9 \cdot 10^2$ | $7.5 \cdot 10^3$ | $-3.1 \cdot 10^3$ |
| | Mass transport (mean mass) [kg/day] | $2.6 \cdot 10^3$ | $-4.7 \cdot 10^2$ | $1.0 \cdot 10^3$ | $-2.8 \cdot 10^2$ | $1.5 \cdot 10^3$ | $-7.5 \cdot 10^2$ |
| | Mass transport (median mass) [kg/day] | $1.4 \cdot 10^3$ | $-2.2 \cdot 10^1$ | $5.7 \cdot 10^2$ | $-6.1 \cdot 10^1$ | $7.6 \cdot 10^2$ | $-2.5 \cdot 10^2$ |
| | River discharge [m³/s] | $3.7 \cdot 10^2$ | $-2.0 \cdot 10^2$ | $4.4 \cdot 10^2$ | $-1.2 \cdot 10^2$ | $5.0 \cdot 10^2$ | $-3.4 \cdot 10^1$ |
| | Flow velocity [m/s] | 0.1 | <0.0 | 0.1 | <0.0 | 0.2 | <0.0 |
| $d_1$ | Item transport | 0.58 | -0.15 | 0.48 | -0.14 | 0.66 | -0.35 |
| | Mass transport (mean mass) | 0.57 | -0.16 | 0.41 | -0.16 | 0.60 | -0.37 |
| | Mass transport (median mass) | 0.62 | -0.02 | 0.48 | -0.08 | 0.61 | -0.28 |
| | River discharge / Flow velocity | 0.32 | -0.25 | 0.39 | -0.15 | 0.44 | -0.04 |
| $d_2$ | Item transport | 1.15 | -0.31 | 0.96 | -0.27 | 1.31 | -0.70 |
| | Mass transport (mean mass) | 1.15 | -0.32 | 0.81 | -0.32 | 1.20 | -0.74 |
| | Mass transport (median mass) | 1.24 | -0.04 | 0.96 | -0.15 | 1.22 | -0.56 |
| | River discharge / Flow velocity | 0.65 | -0.50 | 0.77 | -0.31 | 0.87 | -0.08 |
| $d_3$ | Item transport | 0.73 | -0.27 | 0.65 | -0.24 | 0.79 | -0.52 |
| | Mass transport (mean mass) | 0.73 | -0.27 | 0.58 | -0.28 | 0.75 | -0.54 |
| | Mass transport (median mass) | 0.77 | -0.03 | 0.65 | -0.14 | 0.76 | -0.44 |
| | River discharge / Flow velocity | 0.49 | -0.40 | 0.56 | -0.27 | 0.61 | -0.08 |

**Independent of the selected delivery ratio equations, our main conclusions are still supported. The delivery of plastics is higher than that of water for all $d_1$, $d_2$ and $d_3$ values (Table R1). Ebb dominated cycles have positive delivery values (net downstream transport) whereas flood dominated cycles are characterized by negative delivery values (net upstream transport) (Table R2). Also, our finding that the net plastic transport is limited by tidal dynamics remains correct, as delivery values over the entire monitored period were all found to be below unity. The extent of the reduction in transport is however variable depending on the calculation method chosen. Because we want to constrain delivery values between +1/-1 and 0, we consider $d_2$ to not be a suitable option. We thus suggest to keep using $d_1$. We are interested to hear from you whether we may have overlooked any fundamental shortcomings in the conceptualization of the delivery ratio. If fundamental flaws are highlighted in using $d_1$ (see eq. 1), $d_3$ could be a suitable alternative.**

Third the unnecessary (and silly) addition of 100% in Eq.(6). As the authors should be aware, 100%=1, so there is no need to multiply by 100%. If a delivery value is 18%, then that is the same as 0.18.

**We just wanted to emphasize that we expressed the delivery as a percentage. We will remove this from the equation, as we agree it is an unnecessary addition, and now express delivery values as a fraction.**

Fortunately, it seems as if the equations are wrong, but the calculations are right. A quick look at the calculations in Table 1 suggests that they are correct.

2*3.1/(15+8.6)=0.27

2*170/(1100+790)=0.18

**We did not report in the previously submitted version of Table 1 nor in other parts of the manuscript the volumes we estimated. We reported the transport rates during ebb and flood, as well as net transport rates. We realize this is probably confusing to the reader. We initially did not report those values because literature on river plastic reports concentrations or transport rates. In a revised version of Table 1, which will be similar to Table R1 we will add the volumes as well. For the other tables, due to space limitations, we will not add the volumes. But we will report those in Annex.**

The delivery ratio of the flow velocity is calculated as 0.18. This seems to me a realistic value for the ratio between the freshwater volume and the tidal volume, so it seems as if the calculations are right. In that case the equations provided are wrong and the calculations are correct. But without more detailed insight I cannot conclude either way.

Then the interesting question remains why the plastic has a higher delivery ratio than the fresh water. My hypothesis would be that this has to do with the lateral distribution of the floating plastic. Floating plastic has the tendency to concentrate mid-stream, particularly during ebb (from my own observations in many tidal rivers). In midstream the surface flow velocities are largest. The concentration of floating objects (also water hyacinth) in mid-stream is due to the helix movement of water in river bends where floating objects are brought together. At slack time the water slacks earlier near the banks which causes a lateral movement of the surface water towards the banks. At flood flow, the floating objects are spread over the width and may even be trapped in the banks. The net transport of floating objects is then less. Maybe you can check if your observations confirm this. In any case, the data suggest that the plastic is discharged faster to the ocean than the average (freshwater) flow velocity suggests.

**Thank you for this very interesting remark about the plastic lateral distribution. We have done some additional analysis to explore the plastic and flow velocity lateral distributions, as well as their relationships. Ultimately, we wanted to check: 1) whether the mid-stream segment (e.g.: the segment where the flow velocity is the highest) coincides with the segment where plastic transport peaked, for the ebb and flood phases; 2) whether plastic transport is more widely spread over the width for the flood phase than during the ebb phase. For your reference, we have plotted the lateral distribution of plastic transport and flow velocity per tidal cycle, for ebb and flood (Figure R2). We also estimated the relative plastic transport per segment (expressed in relative contribution to the mean cross-sectional transport rate) (Table R3).**

[Figure]

**Figure R2.** Lateral distribution of plastic transport and flow velocity for the six tidal cycles considered.

**Regarding the first point, for both ebb and flood, in 50% of the tidal cycles the peak flow velocity and plastic transport occurred at the same segment (Table R3). During the ebb phase, plastic transport typically peaked at segment 2 or 3; for the flood phase, at segments 4 or 5. We also calculated the percentage of observation rounds (14 to 23 per tidal cycle) for which the peak flow velocity and plastic transport occurred at the same segment: 67% for ebb and 72% for flood. In addition, we calculated the Spearman's correlation coefficient between flow velocity and plastic transport for all observation rounds. We found low Spearman's rho values (0.35 and 0.32 for ebb and flood, respectively; p-value < 0.01), and no considerable difference between ebb and flood. Note that these results are very different than those presented in our manuscript in Figure C1, Appendix C. In Figure C1 we present the relationship between discharge and plastic transport averaged across the cross-section. In summary, based on our observations, we cannot conclude that plastic transport is more concentrated mid-stream during the ebb phase than during the flood phase.**

**Table R3.** Relative lateral distribution of plastic transport per observation segment with respect to the mean cross-sectional transport

| A. Ebb phase | | | | | | | | B. Flood phase | | | | | | |
| --- | --- | --- | --- | --- | --- | --- | --- | --- | --- | --- | --- | --- | --- | --- |
| **Cycles** | **1** | **2** | **3** | **4** | **5** | **6** | | **Cycles** | **1** | **2** | **3** | **4** | **5** | **6** |
| **Segments** | | | | | | | | **Segments** | | | | | | |
| 1 | 0.19 | 0.09 | 0.12 | 0.03 | 0.41 | 0.26 | | 1 | 0.07 | 0.06 | 0.04 | 0.01 | 0.04 | 0.08 |
| 2 | 0.42 | 0.33 | 0.51 | 0.29 | 0.35 | 0.37 | | 2 | 0.14 | 0.13 | 0.14 | 0.19 | 0.13 | 0.12 |
| 3 | 0.11 | 0.19 | 0.16 | 0.23 | 0.14 | 0.13 | | 3 | 0.22 | 0.03 | 0.11 | 0.05 | 0.08 | 0.27 |
| 4 | 0.12 | 0.22 | 0.13 | 0.24 | 0.07 | 0.13 | | 4 | 0.20 | 0.49 | 0.11 | 0.28 | 0.27 | 0.37 |
| 5 | 0.16 | 0.18 | 0.08 | 0.21 | 0.03 | 0.11 | | 5 | 0.37 | 0.30 | 0.60 | 0.48 | 0.47 | 0.15 |

**For the second point, we investigated the distribution of plastic transport over the cross-section during both flood and ebb phases (Figure R3). We calculated the RMSE for both ebb and flood, and found that plastic transport during the ebb phase is more uniformly distributed than during the flood phase (RMSE: 0.095 and 0.19 for ebb and flood, respectively). This proves the opposite of the initial hypothesis, e.g. that plastic transport is more uniformly distributed across the river width during the flood than the ebb phase.**

[Figure]

**Figure R3.** Cumulative relative distribution of plastic transport across the river width, for ebb (A) and flood (B) tidal phases.

**Considering these elements, we cannot conclude than the lateral distribution of floating plastic transport might explain the higher delivery values for plastic compared to freshwater. In our case, the explanations provided in section 4.3, namely that 1) other factors than flow velocity, such as wind influence floating plastic transport, and 2) entrapment of plastic items downstream of the monitored site could explain the higher delivery ratio values found compared to water. However, we agree that in other cases, the plastic lateral distribution could be a factor affecting the net transport and delivery of plastic transport, thus we will mention this in a revised version of the discussion section.**

In line 310 the authors observe that the travel times of plastic in estuaries is long. This is not surprising since estuaries have an exponential shape (see Savenije, 2012) with cross-sectional areas orders of magnitude larger than the river cross-section. The section of observation in the Saigon River is rather far from the ocean, so the increase of the cross-sectional area is still modest compared to the river width, but further down the widening is much more and hence the delivery ratio will reduce substantially as one moves downstream. Generally, in alluvial estuaries the net downstream velocity is much smaller than that of the river feeding the estuary. So, retention times in estuaries are long and there is nothing special about it. But what is surprising is that the delivery ratio of the plastic is higher than the net freshwater transport. That is an interesting finding!

**Thank you for your remarks. Indeed, there is nothing special about long retention times in estuaries. However, observation-based studies of plastics in estuaries are very scarce and global river plastic models have so far entirely neglected tidal dynamics. Thus we believe it is important to stress these results to the reader. We agree that finding higher delivery of plastics compared to net freshwater is an interesting finding.**

In Section 4.3 the authors discuss this issue. I think the explanation of entrapment (particles getting temporarily stuck during the flood flow) is realistic, but I would like the authors to ponder on the lateral distribution of the floating plastic. The fact that the floating plastics have a higher delivery ratio than submerged plastic supports my idea of the lateral distribution playing a role. I guess that your observations at the bridge with 5 observation sections could be used to investigate this hypothesis.

**See our previous answer on this point.**

In lines 446-447 the authors suggest that it is the tidal movement that hampers the transport of the plastic. This is not true. It is the exponential shape of the estuary that enhances travel times. As one moves downstream, the cross-sectional area increases exponentially and the delivery ratio will decrease proportionally, because the tidal volume is directly proportional to the cross-sectional area (see Savenije, 2012).

**Many thanks for this comment. We agree that the formulation in lines 446-447 is misleading and we will modify it in a revised version of the manuscript.**

**Ultimately, we would like to convey these main points regarding the overall influence of tidal dynamics on plastic transport:**

**1) Plastic is not just emitted from rivers into the ocean. The tidal dynamics impact the transport, increasing retention times, and this increases the likelihood of (temporary) trapping. This has not been included in plastic transport models and emission estimates, and it should be.**

**2) The retention time along the estuary depends on the freshwater discharge, and the estuarine geometry. We could hypothesize that closer to the river mouth, the flow velocities decrease due to a widening of the estuary. This might lead to an increase in deposition of floating plants on the riverbanks. This point requires additional research at different locations within the estuary and was beyond scope for this first study that looked at the impact of tides on plastic transport. We will elaborate this in the discussion with concrete suggestions for future research.**

**Following your comment, we think it would be better to change the manuscript's title. Our submitted manuscript was entitled "Tidal dynamics limit river plastic transport". We plan on changing this title to the following: "River plastic transport affected by tidal dynamics", to better reflect the findings and relevant processes highlighted in our work. In the absence of tides, the net outflow would be equal to the freshwater discharge. Similarly, and without considering deposition and remobilization dynamics, the net volume of plastics being transported in a tidal system would be equal to the volume of plastics transported downstream in a system not affected by tides. There is no 'reduction' in volume due to the tidal prism. However, because of bidirectional flows, water and floating plastics are transported over longer distances for the same period of time compared to a freshwater system of reference. Indeed, the sum of the ebb excursion and the flood excursion tidal excursion is almost six times higher than the difference between ebb and flood excursion (representing the integral of flow velocity) (Table R1). Lotcheris et al. (2023) tracked floating plastic trajectories in the Saigon river, a few weeks after the measurements presented in this manuscript, for the same measurement location. They used a Lagrangian approach, with GPS trackers mimicking the trajectories of floating macroplastics at the river surface. They found that the total distance travelled by plastics is on average ~2 times higher than the net plastic travelled distance. The lower difference between net and total travelled distances found for plastics than for water (2 versus 6) also highlights the frequent deposition of plastics. Frequent stopping and deposition of floating plastics were directly observed in observational studies such as Lotcheris et al., Newbould et al. (2021), Duncan et al. (2020), Mani et al. (2023), Tramoy et al. (2020).**

**The longer distances travelled by plastics in a tidal system compared to a non-tidal one mean that items have a higher likelihood of interacting with the river environment. This could lead to deposition mechanisms, as also observed in Lotcheris et al. (2023). However, changes in water level and flow velocity could also lead to the re-mobilization of plastic items that were previously deposited at river infrastructure, in floating or riparian vegetation and on riverbanks. In that sense, the delivery values do not just give information on the tidal rivers capacity to transport items, but could also indicate changes in the amounts of transported plastics between the ebb and flood phase. In our analysis, we found higher delivery values of plastics compared to water for almost all tidal cycles (with the exception of the fourth cycle, see Table R2). The higher delivery ratio for plastics compared to water could be caused by either deposition of items downstream of the measurement location, or re-mobilization of items upstream of the measurement location.**

**In summary, we will nuance our initial statement that tidal dynamics limit river plastic transport and rephrase it in a revised version of the manuscript. We suggest to reformulate as follows:**

**1) Plastics in tidal rivers travel for longer distances than in non-tidal systems. With longer travelled distances, floating plastics are more likely to interact with the river environment, get deposited and re-mobilized compared to plastics transported in a freshwater system. Considering a fixed river domain, changes in plastic amounts during ebb and flood phases are expected due to deposition and re-mobilization mechanisms, which have been observed in a wide range of river systems.**

**2) Our observations highlight these changes in plastic amounts during ebb and flood phases (caused by deposition and re-mobilization), as we found higher delivery ratios for plastics than water. Fundamental differences in plastic transport compared to water movement (for instance wind influence) could partially explain the higher delivery ratio found for plastics than for water. However, the significant and consistent differences observed strongly suggest that the transported amounts of plastics change throughout the tidal cycles. Further studies could seek to quantify the likelihood of deposition and re-mobilization of floating plastics, in order to account for changes in plastic quantities in transport over tidal cycles.**

Minor observations:

Line 200. The "total river surface length" (the integral of the tidal velocity) is called the "tidal excursion". The ebb excursion is larger than the flood excursion, the difference being the integral of the freshwater velocity. As one moves further downstream the difference between these two excursions becomes smaller until, near the estuary mouth, the difference can hardly be observed anymore.

**Thank you for noticing this. This is well noted and we will correct this in a revised version of the manuscript.**

Reference:

Savenije, H.H.G., 2005, 2012. Salinity and Tides in Alluvial Estuaries, Elsevier. Completely revised 2nd edition in 2012, available from www.salinityandtides.org.

**References in this rebuttal:**

Duncan, E.M., Davies, A., Brooks, A., Chowdhury, G.W., Godley, B.J., Jambeck, J., Maddalene, T., Napper, I., Nelms, S.E., Rackstraw, C., Koldewey, H., 2020. Message in a bottle: Open source technology to track the movement of plastic pollution. PLOS ONE 15, e0242459. doi:10.1371/journal.pone.0242459. publisher: Public Library of Science (PLoS).

Lotcheris, R., Schreyers, L., and van Emmerik, T., 2023. Plastic does not simply flow into the sea: River transport dynamics affected by tides and floating plants. Submitted to Environmental Pollution. https://papers.ssrn.com/sol3/papers.cfm?abstract_id=4449742

Mani, T., Hawangchu, Y., Khamdahsag, P., Lohwacharin, J., Phihusut, D., Arsiranant, I., Junchompoo, C., Piemjaiswang, R., 2023. Gaining new insights into macroplastic transport 'hotlines' and fine-scale retention-remobilisation using small floating high-resolution satellite drifters in the Chao Phraya river estuary of Bangkok. Environmental Pollution 320,121124. doi:10.1016/j.envpol.2023.121124.

Newbould R.A., Powell D.M. and Whelan M.J., 2021. Macroplastic Debris Transfer in Rivers: A Travel Distance Approach. Frontiers in Water 3:724596. doi: 10.3389/frwa.2021.724596

Tramoy, R., Gasperi, J., Colasse, L., Silvestre, M., Dubois, P., Nous, C., Tassin, B., 2020. Transfer dynamics of macroplastics in estuaries – New insights from the Seine estuary: Part 2. Short-term dynamics based on GPS-trackers. Marine Pollution Bulletin 160, 111566. doi:10.1016/j.marpolbul.2020.111566. publisher: Elsevier BV.

---

## Author Comment (AC2)

This paper presents observations of plastic transport in relation to flow velocity and discharge in the tidal part of the Saigon river. It is an interesting, well-written paper with novel observations of plastic transport. I will refrain from the interesting points that Reviewer 1 mentioned and the follow-up response by the authors and focus more on the methods of this research.

Thank you very much for your constructive and critical comments which will improve the quality of our manuscript.

1. Regarding the monitoring of plastics, in line 115 and in Section 2.3 (and in other parts) you mention that the counting of the plastic particles was done visually. How exactly? From the top of the bridge that is 14 m above the water or the plastics were somehow collected and sampled? While reading, in the beginning I assumed the former (which would then lead to the obvious question how accurate this data monitoring is when observing plastics as small as 0.5 cm) but when you mentioned the mass of the plastics in Equation (2) I assumed that you collected the plastics to weigh them. Then, based on the lines 231-235, did you actually collect and classify the plastic samples or did you visually observe them from the bridge and used the distributions from van Emmerik et al. (2019)? This part (and the Section 3.3) is very confusing, please clarify how you sampled the plastics and what is the role of the data from van Emmerik et al. (2019). This part is also critical for the interpretation of Figure 4 and for the analysis related to the different categories of plastics.

Thank you very much for this comment. We will amend section 2.3 to clarify which method we used to monitor floating macroplastic. We used the visual counting method, a widely used measurement method in macroplastic research (González-Fernández et al. 2021; Castro-Jiménez et al., 2019; van Emmerik et al., 2022a; Sarminingsih et al., 2022). Trained observers stand on the bridge and count all visible items over a predefined time interval. Given the spatial heterogeneity in the distribution of floating macroplastics across the river width, this counting process is typically conducted at multiple locations across the bridge. The visual counting method does not necessitate the collection of macroplastics through means like nets. Determining the minimum detectable size of floating items is challenging, as it varies depending on observer sight and perception; ambient conditions and the bridge height. Nonetheless, it is generally accepted that floating items above 0.5 cm are observable (van Emmerik et al., 2018; Liro et al., 2020).

The visual counting method enables one to estimate item transport rate, quantified as number of items per unit of time. It is common in plastic research studies to also report transport rates in terms of plastic mass per unit of time. We thus converted our estimates of item transport rates into mass transport rates. This conversion was done using the dataset on plastic item mass as presented in the work of van Emmerik et al. (2019).

1. In my opinion, for an experimental study there are many assumptions when processing the data and many speculations when interpreting the results, which make the

conclusions a bit doubtful. One of the key findings of the paper, according to the authors, is that the net transport of plastics is higher than the net water discharge. However, the water discharge was estimated based solely on near-surface velocity measurements in a flow environment that can potentially be very complex. The authors multiplied the near-surface measurements with a coefficient 0.85 (please provide proper referencing for this), which I am assuming corresponds to a fully developed boundary layer that obeys the law of the wall. However, the measurements are done in the vicinity of a bridge (actually they included the effect of the bridge by measuring after the flow passed the bridge by changing measuring locations during ebb and flood – in line 109, what does "face the flow direction during measurements" imply?), where local flow accelerations may take place and/or local variations on the bed level may be present with the development of scour holes. In addition, the interactions of fresh and salt water are completely neglected and it is assumed that there is no stratification or mixing that could affect the law of the wall. Finally, by estimating the flow discharge with this method, the (limited time period of) tidal reversal cannot be properly taken into account. As a result, the calculation of the flow discharge is questionable; however, it is used to deduce one of the main findings of the paper: the fact that the net transport of plastics is higher than the net discharge.

Thank you for this comment. We provide clarifications to your comments on the following aspects:

1. **Use of a coefficient of 0.85 to compute averaged-depth flow velocity from near-surface flow velocity measurements**.

We indeed used a coefficient of 0.85 for converting near-surface flow velocity measurements to averaged-depth flow velocity. This coefficient assumes a logarithmic vertical velocity distribution and a typical bed roughness and is generally accepted in the hydrological community (Muste et al., 2008; Boiten, 2003). Haut et al. (2018) estimated averaged-depth flow velocity using gauging data at 176 sites, combining surface flow velocity measurements with water level data and found that most coefficient values fall between the range of 0.7 and 0.9. We will update our referencing to include this study and others regarding the use of a coefficient of 0.85.

Additional references that will be included in a revised version of the manuscript:

Boiten, W. (2003). *Hydrometry: IHE Delft lecture note series*. CRC press.

Hauet, A., Morlot, T., & Daubagnan, L. (2018). Velocity profile and depth-averaged to surface velocity in natural streams: A review over a large sample of rivers. In *E3s web of conferences* (Vol. 40, p. 06015). EDP Sciences.

Muste, M., Fujita, I., & Hauet, A. (2008). Large-scale particle image velocimetry for measurements in riverine environments. *Water resources research*, *44*(4).

Rantz, S. E. (1982). Measurement and computation of streamflow. USGS Water-Supply Paper 2175. *US Geological Survey, Reston, VA*.

**2. Small-scale processes that might influence discharge estimates (scour holes, local variations in bed level, fresh and saltwater mixing)**

We acknowledge that bathymetric data collected at the monitoring site could provide more accurate estimates of water depths and potentially reveal local scour holes and local variations in the riverbed. These data are however not available. Nevertheless, we measured water depths at five locations across the river width, taking into account contraction scour effects (Arneson, 2013). However, we did not directly measure water depths at the nose of the bridge piers, which could mean that we may have overlooked local scour holes.

To address this, we here estimate the impact of local scour holes on our cross-sectional area calculations and, consequently, on our discharge estimates. We have taken a conservative approach, assuming a worst-case scenario for our calculations. We estimated the maximum scour hole depths for the considered bridge piers. The methodology for estimating these depths can be found in Arneson's work (2013) in Chapter 7, specifically detailed in equations 7.3 and 7.4.

Local scour hole depth ($y$) [m] was calculated at each point $i$ in front of bridge piers, using the following equation:

$$y_i = 2.0 \, K_1 K_2 K_3 \left(\frac{h_i}{a}\right)^{0.35} Fr_i^{0.43} \, a$$

where:

$y$ = Scour depth [m]
$h$ = Flow depth directly upstream of the pier [m]
$K_1$ = Correction factor for pier nose shape
$K_2$ = Correction factor for angle of attack of flow
$K_3$ = Correction factor for bed condition
$a$ = Pier width [m]
$Fr$ = Froude number directly upstream of the pier $= \dfrac{v_i}{(h_i g)^{0.5}}$

$v$ = Mean velocity of flow directly upstream of the pier [m/s]
$g$ = Acceleration of gravity [9.81 m/s]

We calculated the Froude number ($Fr$), flow depth ($h$) and flow velocity ($v$) for each bridge pier $i$ that we considered. Note that the bridge pier closest to the north-west riverbank was excluded from this analysis. This decision was based on its close proximity to the bank; in an area characterized by very low flow velocities and shallow water depths. These conditions make it unlikely for scour holes to form. Flow depths and flow velocities in front of bridge

piers were estimated by averaging the flow depths and flow velocities measured at each observation point. We acknowledge that this is a simplification, as there were instances where the bridge piers were nearer to one of the two observation points.

[Figure]

**Figure R1.** Bridge piers considered, and representation of flow velocity ($v$) and water depths ($h$) upstream of each pier $i$. The bridge pier marked with a red cross was not considered as likely to have scour holes. Copyright: Bing Maps.

The correction factor for angle of attack of flow ($K_2$) was calculated as follows:

$$K_2 = \left(cos\,\theta + \frac{L}{a}\,sin\,\theta\right)^{0.65}$$

$\theta$ = Angle of attack of the flow [degrees]
$L$ = Length of the pier [m]

The bridge piers at the monitored location are cylinders of diameter of approximately 2.4 m ($L = a = 2.4\ m$). The angle of attack of the flow was estimated at 12 degrees, using aerial imagery. As a result, $K_2$ was estimated at 1.17. Additionally, we assigned a value of 1.0 to $K_1$, given the round nose of the piers. $K_3$ was set at 1.1, considering the possibility of a bed condition with small dunes (Arneson, 2013).

We found scour depths reaching maximum values between 3.6-2.7 m, depending on the bridge pier considered. To estimate the scour hole area around each bridge pier we use the following equation:

$$A_{s,i} = y_i\,b_i$$

where $b$ indicates the topwidth [m] of the scour hole. Note that we consider the scour hole area around each pier to be comprised of two triangular areas on either side of the pier. Arneson (2013) recommends estimating the topwidth $b$ as twice the scour depths (i.e., $b$ = 2y).

In the end, the total scour area across the entire cross-section amounts to approximately 90 m$^2$. This results in an increase in river discharge estimates of 2%. Thus, we can reasonably assume that under such worst-case scenario, factors such as local scour holes have only a minimal impact on our discharge estimates.

In addition, it should be noted that precise quantification of discharge was outside the scope for our study. We estimated river discharge to enable comparisons between water flow and plastic transport dynamics. Furthermore, in our comparisons of water delivery ratios to that of plastics, we incorporate both flow velocity and discharge-based estimates. As you rightly noted in a further comment, using flow velocity estimates results in lower uncertainty.

**3. The influence of the bridge**

Thank you very much for mentioning the influence of the bridge. In the next sub-section ('Tidal reversal') we expand on the influence of the bridge with respect to tidal reversal and the flow direction. Here we focus on the influence of the bridge on the flow dynamics. The presence of the bridge piers indeed obstructs a portion of the flow and the transport of plastics. We conducted our measurements at observation points with minimal direct disturbance from the bridge piers (see Figure R2 below). Consequently, we assume that our measurements are representative for the natural undisturbed river cross-section.

[Figure]

**Figure R2**. Measurement site (Thu Thiem bridge, 10.785984, 106.718332) and locations. The numbers 1, 2, 3... mark the observation points distributed across the bridge, with variations in their location depending on the flow direction. For floating plastic, we considered observational track width $w_i$ (of 15 meters). For discharge calculations, we considered widths represented as $S_i$ at each observation point. Copyright: Bing Maps.

**4. Tidal reversal**

Tidal reversal was taken into account in our estimates, as we measured continuously near-surface flow velocity and water depths throughout six tidal cycles. One measurement (including flow velocity, water depth and plastic transport rates) took an average of 9 minutes.

We indeed conducted our measurements facing the flow, so when water (and plastics) had already passed beneath the bridge (see Figure R2). This approach was necessary, particularly for the measurements of surface flow velocity and water depths, as it allowed us to handle the equipment with care. Conducting these measurements prior to water passing beneath the bridge would have led to a loss of visibility of the equipment, as it would have been positioned beneath the bridge due to the flow.

2. Moreover, the way Equation (4) is written, implies that it doesn't calculate the discharge of the whole cross section. In line 133 you mention that w_i=15 m (and W=298 m) and you only have 5 such widths. So, by summing these five areas, you estimate a partial discharge of the river. This is not necessarily a problem, but you relate this to the plastic transport, F, in Equation (1), which extrapolates the measured plastics transport from each width w_i to the whole river width W. By measuring so close to the bridge, it is expected that the bridge piers will induce a high variation in the flow velocities and the plastic concentration on the water surface across the river cross-section. Please clarify how these variables are connected.

Thank you for this comment. We acknowledge that the equations used to estimate discharge imply a partial discharge calculation. For floating plastic, we considered an observational track width of 15 m (represented as $w_i$ in the figure below). For discharge calculations, we considered widths represented as $S_i$ at each observation point.

We propose to amend equation 3 to explicitly indicate that we calculated the discharge over the entire cross-section, rather than partial discharge:

$$a_i = S_i \cdot d_i$$

Furthermore, we have made revisions to Figure R2 to illustrate both " $w_i$ " and " $S_i$ " for improved visual representation. We will update Figure 1B of the manuscript as shown in Figure R2. Note that the bridge decks visible from the aerial imagery between observation points 2 and 3, and between 4 and 5, are much larger in size than the bridge piers.

[Figure]

**Figure R2**. Measurement site (Thu Thiem bridge, 10.785984, 106.718332) and locations. The numbers 1, 2, 3... mark the observation points distributed across the bridge, with variations in their location depending on the flow direction. For floating plastic, we considered observational track width $w_i$ (of 15 meters). For discharge calculations, we considered widths represented as $S_i$ at each observation point. Copyright: Bing Maps.

Some other comments in order of appearance:

3. Lines 142-143: There is no justification about this assumption for the categories "Multilayer" and "Other plastic" and the choices seem rather arbitrary.

We understand your concern regarding the assumption made for the "Multilayer" and "Other plastic" categories. Indeed, there were no measurements of plastic mass available for these two categories. Considering that the 'Multilayer' items were previously categorized as 'PO soft' items in the study conducted by van Emmerik et al. in 2019, we suggest lumping together 'Multilayer' and 'PO soft' categories for the mass estimates. For 'Other plastic' we keep the overall mean of all found macroplastic items. In a revised version of the manuscript, we will update our results on mass transport estimates.

4. Lines 168-170: I agree with the authors that it's less uncertain discussing flow velocities that are directly measured instead of the calculated flow discharges; however, a large part of the analysis is still done based on water discharges and water volumes. How reliable are your conclusions then?

We report both flow velocity and discharge values, and we base our conclusions on both. We obtained similar results using both flow velocity and discharge, which enhances the robustness and confidence in the study's conclusions.

5. Line 201: In the way that you defined the plastic transport (either items per hour in Equation 1 or mass per day in Equation 2), how can you get volume of plastic transport by integration?

We acknowledge that the phrasing at l. 201 is unclear and propose to revise this paragraph accordingly:

*"The integral values for flow velocity and discharge correspond respectively to the total river surface length [m] and river volume [m³] that passed by the measurement location per tidal phase. The integral values for plastic transport corresponds to the total amount and mass of plastic items passing by the measurement location."*

6. Line 218: Why only higher net transport of plastics and not lower?

We appreciate your valuable input. The current wording of line 218 is indeed in need of revision. It's important to note that multiple factors can contribute to the observed differences in plastic and water delivery ratios, which could lead to either higher or lower net transport of plastics compared to water. We propose the following rephrased version:

*"Over the six tidal cycles considered, we found a seaward mean net transport of approximately $3.1 \cdot 10^3$ items hour$^{-1}$, corresponding to 400-760 plastic kg day$^{-1}$ (Table 1). This represents only about 27-32% of total plastic transport. This ratio is lower for river discharge and flow velocity (18%) (Table 1). In the Discussion, we explored potential explanations for the observed disparities between water and plastic delivery ratios".*

7. Line 224: Can you please clarify "with both higher peaks in flow velocity during the ebb than flood phase of the tidal cycles"

The current wording of line 224 needs rephrasing. We propose the following rephrased version:

*"The maximum flow velocity during the ebb phase exceeds that observed during the flood phase (0.56 and -0.41 m s$^{-1}$, respectively)".*

8. Lines 278-279: What are the three values and how is this related to the next sentence and the median mass of items?

We agree that the current wording at l.278-9 needs rephrasing. We suggest the following:

*"Overall, calculated plastic delivery ratios based on items transport and mass transport are in good agreement, with no more than ±5% of difference between the calculated values (item transport, mass transport based on median, and mean mass per item)."*

9. Lines 337 and 404: As a reader, a paper that is under review does not provide strong support for an argument.

We'd like to clarify that since the time of your review, the reference in line 337 has been published. We will update the reference accordingly to:

Tasseron, P., Begemann, F., Joosse, N., van der Ploeg, M., van Driel, J. and van Emmerik, T. (2023). Amsterdam Urban Water System as an Entry Point of River Plastic Pollution. *Environmental Science and Pollution Research* 30, 73590–73599.

Regarding the manuscript mentioned in line 404 (Lotcheris et al., 2023 in review), while it has not been published yet, it represents a field-based study focusing on the Saigon river and tidal dynamics. Given its relevance to our work, we believe it is appropriate to include a reference to this study in our manuscript.

Lotcheris, R., Schreyers, L., Bui, T. K. L., Thi, K., Nguyen, H. Q., Vermeulen, B., & van Emmerik, T. Plastic Does Not Simply Flow into the Sea: River Transport Dynamics Affected by Tides and Floating Plants. *Available at SSRN:* https://ssrn.com/abstract=4449742

10. Lines 356-357: Is this reference at the tidal part of the river? If not, how is it related to your study?

We greatly appreciate your observation on lines 356-357. We aim to reference studies that specifically address the tidal regions of rivers to maintain relevance to our research focus. We propose to amend lines 356-357 as follows:

*"The large discrepancy between instantaneous and net plastic transport highlights the need to estimate transport rates based on longer observation periods than usually done in riverine transport studies. For example, González-Fernández et al. (2021) quantified plastic transport over 42 rivers, including 5 influenced by tides. Similarly, van Emmerik et al. (2022a) estimated plastic transport in Dutch rivers, encompassing 26 locations, 7 of which were influenced by tides. In both studies, data collection was limited to the ebb phase, which may have led to potential overestimations of plastic transport."*

11. Lines 392-394: This is a very vague and highly speculative sentence.

Thank you for this remark. We agree that the last two sentences of that paragraph are quite speculative, and thus suggest to remove them. We propose the following amended paragraph:

> *"Besides flow velocity and discharge, other factors could influence the velocity of plastics, such as wind, waves and lateral flows (Laxague et al., 2018; van der Mheen et al., 2020). These factors could generate accelerating or decelerating effects in the propagation of plastic in the river".*

**Reference list from this reply:**

Arneson, L. A. (2013). *Evaluating scour at bridges* (No. FHWA-HIF-12-003). United States. Federal Highway Administration.

Boiten, W. (2003). *Hydrometry: IHE Delft lecture note series*. CRC press.

Castro-Jiménez, J., González-Fernández, D., Fornier, M., Schmidt, N., & Sempéré, R. (2019). Macro-litter in surface waters from the Rhone River: Plastic pollution and loading to the NW Mediterranean Sea. *Marine Pollution Bulletin*, *146*, 60-66.

González-Fernández, D., Cózar, A., Hanke, G., Viejo, J., Morales-Caselles, C., Bakiu, R., ... & Tourgeli, M. (2021). Floating macrolitter leaked from Europe into the ocean. *Nature Sustainability*, *4*(6), 474-483.

Hauet, A., Morlot, T., & Daubagnan, L. (2018). Velocity profile and depth-averaged to surface velocity in natural streams: A review over a large sample of rivers. In *E3s web of conferences* (Vol. 40, p. 06015). EDP Sciences.

Laxague, N. J., Özgökmen, T. M., Haus, B. K., Novelli, G., Shcherbina, A., Sutherland, P., ... & Molemaker, J. (2018). Observations of near-surface current shear help describe oceanic oil and plastic transport. *Geophysical Research Letters*, *45*(1), 245-249.

Lotcheris, R., Schreyers, L., Bui, T. K. L., Thi, K., Nguyen, H. Q., Vermeulen, B., & van Emmerik, T. Plastic Does Not Simply Flow into the Sea: River Transport Dynamics Affected by Tides and Floating Plants. *Available at SSRN*: https://ssrn.com/abstract=4449742

Liro, M., Emmerik, T. V., Wyżga, B., Liro, J., & Mikuś, P. (2020). Macroplastic storage and remobilization in rivers. *Water*, *12*(7), 2055.

Muste, M., Fujita, I., & Hauet, A. (2008). Large-scale particle image velocimetry for measurements in riverine environments. *Water resources research*, *44*(4).

Rantz, S. E. (1982). Measurement and computation of streamflow. USGS Water-Supply Paper 2175. *US Geological Survey, Reston, VA*.

Sarminingsih, A., Andarani, P. and Nugraha, W.D. (2022). Developing a Visual Counting Method to Quantify Riverine Plastic Litter: A Case Study of Rivers in Semarang City, Indonesia. In *IOP Conference Series: Earth and Environmental Science* (Vol. 1098, No. 1, p. 012050). IOP Publishing.

Tasseron, P., Begemann, F., Joosse, N., van der Ploeg, M., van Driel, J. and van Emmerik, T. (2023). Amsterdam Urban Water System as an Entry Point of River Plastic Pollution. *Environmental Science and Pollution Research* 30, 73590–73599.

van der Mheen, M., Van Sebille, E., & Pattiaratchi, C. (2020). Beaching patterns of plastic debris along the Indian Ocean rim. *Ocean Science*, *16*(5), 1317-1336.

van Emmerik, T., Kieu-Le, T. C., Loozen, M., Van Oeveren, K., Strady, E., Bui, X. T., ... & Tassin, B. (2018). A methodology to characterize riverine macroplastic emission into the ocean. *Frontiers in Marine Science*, *5*, 372.

van Emmerik, T., Strady, E., Kieu-Le, T. C., Nguyen, L., & Gratiot, N. (2019). Seasonality of riverine macroplastic transport. *Scientific reports*, *9*(1), 13549.

van Emmerik, T., de Lange, S., Frings, R., Schreyers, L., Aalderink, H., Leusink, J., ... & Vriend, P. (2022). Hydrology as a driver of floating river plastic transport. *Earth's Future*, *10*(8), e2022EF002811.